# Disentangled Causal Transformer: Counterfactual Prediction under Time-Varying Treatments

## Abstract

Estimating longitudinal counterfactual outcomes from observational data is pivotal to personalized medicine and other domains. However, prevailing approaches for mitigating time-varying confounding bias typically balance all covariates indiscriminately, conflating confounders with instrumental variables and thus unnecessarily discarding valuable outcome-relevant information. While causal disentangled representation learning has proven effective in static settings, extending it to the longitudinal setting—where representation disentanglement and time-series modeling must be performed jointly over time—remains a key challenge. To address this, we introduce the **Disentangled Causal Transformer (DCT)**, a Transformer-based architecture designed to integrate causal representation disentanglement seamlessly within the sequence modeling process for robust longitudinal causal inference. DCT features a novel **disentangled multi-head attention** mechanism that decomposes a patient's history into instrumental, outcome, and confounder components. This design enables unbiased causal estimates while preserving the full predictive signal, thus mitigating the traditional trade-off between factual and counterfactual prediction accuracy. Extensive experiments on fully synthetic and semi-synthetic datasets derived from real electronic health records show that DCT consistently outperforms state-of-the-art baselines by a large margin in counterfactual outcome prediction. To the best of our knowledge, DCT pioneers the integration of causal representation disentanglement within a Transformer-based model for robust longitudinal causal inference.

## 1 Introduction

Estimating counterfactual outcomes from observational data is a cornerstone of modern data-driven science, with profound implications for fields like personalized medicine and public health Robins et al. (2000). While randomized controlled trials (RCTs) are the gold standard for causal inference, their practical and ethical limitations necessitate robust methods for causal analysis of observational data Frauen et al. (2024). The widespread adoption of Electronic Health Records (EHRs) provides an unprecedented opportunity, offering rich longitudinal data that chronicle patient journeys Allam et al. (2021). These records enable large-scale causal studies that are often infeasible in traditional clinical trials Hamburg & Collins (2010), particularly in complex, high-stakes domains like oncology, where clinicians must make sequential treatment decisions based on a patient's evolving health state Geng et al. (2017).

While methods for static (cross-sectional) causal inference are well-established Wu et al. (2023); Hassanpour & Greiner (2019b); Cheng et al. (2022); Shi et al. (2019), the longitudinal setting introduces a more formidable challenge: time-varying confounding Robins & Hernan (2008); Bica et al. (2020). In this setting, past treatments influence future covariates, which in turn guide subsequent treatment decisions, creating a dynamic feedback loop that complicates causal analysis. To navigate this complexity, two dominant paradigms have emerged. The first employs re-weighting techniques. Exemplified by Marginal Structural Models (MSMs) with Inverse Probability of Treatment Weighting (IPTW), this approach aims to create a pseudo-population where the causal link between covariates and treatment assignment is effectively severed Robins et al. (2000); Mansournia et al. (2012); Austin & Stuart (2015). The second paradigm leverages representation learning

to achieve covariate balance. Models like the Counterfactual Recurrent Network (CRN) Bica et al. (2020) and the Causal Transformer (CT) Melnychuk et al. (2022) use adversarial training to learn representations of patient history that are invariant to treatment, thereby enforcing independence between the learned state and the treatment administered.

There is a critical flawed assumption underlies both paradigms: they treat all observed pretreatment covariates as confounders.However, a lot of work in static settings shows that the learned representations cannot and should not remove all selection bias Hassanpour & Greiner (2019a); Berrevoets et al. (2021); Cheng et al. (2022). This is because confounders not only contribute to the treatment assignment but also to determining the respective outcomes. Consequently, indiscriminately balancing all covariates can unintentionally remove outcome-relevant information that is intertwined with confounders, leading to over-adjustment and potentially undermining the precision and validity of causal effect estimates.

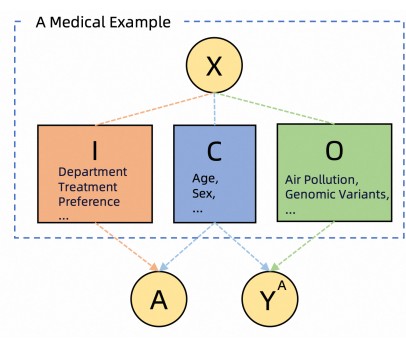

Figure 1: Underlying factors of covariates $X$; $I$ are instruments that only determine treatment $A$; $O$ are outcome factors that only determine outcome $Y$; and $C$ are confounders.

To overcome these limitations, we introduce the **D**isentangled **C**ausal **T**ransformer (DCT), a novel architecture guided by the principles of causal disentanglement Hassanpour & Greiner (2019b); Wu et al. (2023); Cheng et al. (2022). It disentangles the representations of the patient's history into three distinct factors: instrumental factors, outcome factors, and confounders, as shown in Fig. 1. Our key insight is that the multi-head attention mechanism can be repurposed to perform this causal disentanglement directly within the temporal learning process. Specifically, DCT constrains distinct groups of attention heads to operate in separate latent subspaces, each dedicated to one causal factor. This architectural innovation enables the model to dynamically separate instrumental, confounding, and outcome signals as they evolve, rather than treating disentanglement as a post-hoc operation. By doing so, DCT mitigates information loss from over-adjustment and preserves crucial outcome-predictive signals, thereby improving both factual and counterfactual prediction.

In summary, our main contributions are threefold:

- We introduce the Disentangled Causal Transformer (DCT), the first end-to-end architecture to integrate a causal disentanglement framework within a Transformer.
- We design a novel disentangled multi-head attention mechanism that achieves this causal representation disentanglement, learning to disentangle a patient's history into distinct instrumental, outcome, and confounders within independent attentional subspaces.
- We demonstrate through extensive experiments on a fully synthetic dataset and a semi-synthetic benchmark generated from a real-world clinical database that DCT achieves state-of-the-art performance, consistently and significantly outperforming existing methods.

## 2 RELATED WORKS

Our work involves three research domains: counterfactual outcome prediction under time-varying confounding bias, disentangled representation learning, and the application of Transformer architectures to causal inference.

**Counterfactual Outcome Prediction with Time-Varying Confounding**: Early work in epidemiology established principled frameworks for handling time-varying confounding bias, including the g-computation formula, Structural Nested Models (SNMs), and Marginal Structural Models (MSMs) Robins et al. (2000); Robins (1986; 1994). These methods typically rely on linear or logistic models, limiting their capacity to capture complex information in patient trajectories Hernán et al. (2001); Bica et al. (2020). To overcome these expressivity limitations, subsequent research turned to Bayesian non-parametric approaches, for instance, Xu et al. (2016) integrated Gaussian Processes with the g-computation formula, and Soleimani et al. (2017) extended this work using state-space models. The advent of deep learning offered a promising avenue to address these limitations by

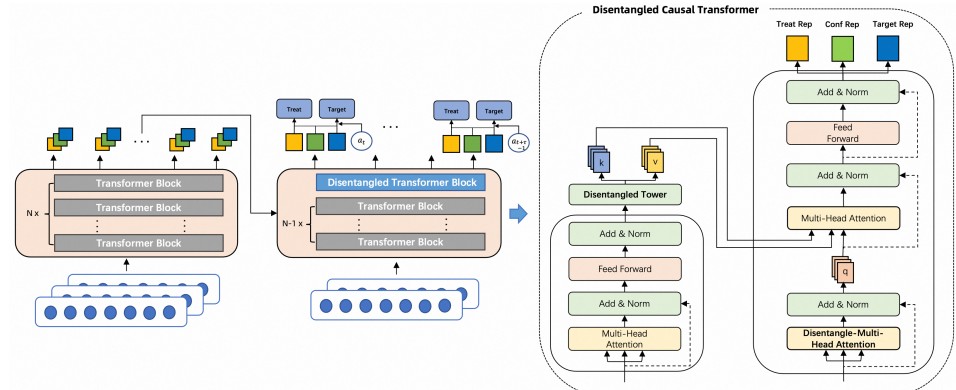

Figure 2: **(Left)** The overall encoder-decoder architecture of our proposed DCT. **(Right)** A detailed view of our proposed Disentangled Causal Transformer. The model processes three types of time-series inputs: outcomes (e.g., diastolic blood pressure), covariates (e.g., age), and treatment interventions (e.g., chemotherapy). The DCT then produces three disentangled representations for **treatment**, **confounders**, and **outcomes**.

allowing for more flexible, data-driven modeling of complex patient dynamics. Recurrent Marginal Structural Networks (RMSNs) Lim (2018) leveraged sequence-to-sequence models for longitudinal data. A prominent strategy within this deep learning paradigm is representation balancing, which aims to align the latent distributions of treated and control groups to achieve covariate balance. Models like the Counterfactual Recurrent Network (CRN) Bica et al. (2020) and the Causal Transformer (CT) Melnychuk et al. (2022) implement this through adversarial training or specialized discrepancy losses to learn a treatment-invariant representation.

**Causal Representation Disentanglement**: The principle of causal representation disentanglement offers a principled solution to the challenges of indiscriminate balancing. Rooted in the understanding that different covariates play distinct causal roles (e.g., confounders, instrumental variables, outcome factors), the objective is to explicitly factor the patient history into these separate causal components Hassanpour & Greiner (2019b); Cheng et al. (2022); Wu et al. (2023). This allows mitigating bias without sacrificing outcome-relevant information that might be erroneously removed by general balancing approaches. Berrevoets et al. (2021) firstly attempted to apply this principle to longitudinal data with the Disentangled Causal Recurrent Network (DCRN). However, DCRN is constrained by two critical limitations: first, its Recurrent Neural Network (RNN) backbone struggles to capture long-range temporal dependencies, and second, its design follows a two-stage process where disentanglement is performed as a post-hoc step on the encoded representation, rather than being integrated into the sequence modeling itself.

**Transformers for Causal Inference**: The Transformer architecture has emerged as the state-of-the-art for modeling long-range dependencies in sequential data Vaswani et al. (2017), revolutionizing fields from Natural Language Processing Devlin et al. (2019); Brown et al. (2020); Bai et al. (2023) to computer vision Dosovitskiy et al. (2020); Liu et al. (2021); He et al. (2022). Despite the parallel successes of Transformers in time-series modeling and causal representation disentanglement in static-setting counterfactual prediction, their synthesis remains a critical, unexplored research gap. Specifically, the question of how to embed the principles of causal disentanglement directly within the Transformer architecture has not been addressed. This gap is particularly significant for longitudinal counterfactual prediction, where the dual requirements of modeling long-range dependencies and achieving precise causal adjustment are paramount.

## 3 PROBLEM DEFINITION

We consider the standard framework for counterfactual outcome estimation in longitudinal observational settings Bica et al. (2020); Feuerriegel et al. (2024); Berrevoets et al. (2021). For each patient, the longitudinal trajectory up to time $t$ is defined by the observed history $\bar{\mathbf{H}}_t = \{\bar{\mathbf{X}}_t, \bar{\mathbf{A}}_{t-1}, \bar{\mathbf{Y}}_t, \mathbf{V}\}$, where $\bar{\mathbf{X}}_t = (\mathbf{X}_1, \ldots, \mathbf{X}_t)$ are time-varying covariates, $\bar{\mathbf{A}}_{t-1} = (\mathbf{A}_1, \ldots, \mathbf{A}_{t-1})$ are past treatments, $\bar{\mathbf{Y}}_t = (\mathbf{Y}_1, \ldots, \mathbf{Y}_t)$ are past outcomes, and $\mathbf{V}$ denotes static features. This dynamic inter-

play of variables, where treatments and covariates influence subsequent outcomes and treatments, inherently leads to time-varying confounding.

Our primary objective is to estimate the expected potential outcome $\tau$ steps into the future, given a hypothetical treatment sequence $\bar{\mathbf{a}}_{t:t+\tau-1} = (\mathbf{a}_t, \ldots, \mathbf{a}_{t+\tau-1})$ and the observed history $\bar{\mathbf{H}}_t$. Formally, we aim to model:

$$\mathbb{E}\left[\mathbf{Y}_{t+\tau}(\bar{\mathbf{a}}_{t:t+\tau-1}) \mid \bar{\mathbf{H}}_t\right] \tag{1}$$

Consistent with Bica et al. (2020); Melnychuk et al. (2022); Lim (2018), we identify this estimand from observational data under the standard assumptions of consistency, positivity, and sequential ignorability (formal definitions in Appendix A.3).

Drawing inspiration from causal disentanglement, we hypothesize that the history $\mathcal{H}_t$ can be encoded into three distinct latent representations:

**Instrumental factors**, $z_t^{\mathrm{I}}$, which influence treatment assignment but are independent of outcome.

**Outcome factors**, $z_t^{\mathrm{O}}$, which influence the outcome but not the treatment assignment.

**Confounding factors**, $z_t^{\mathrm{C}}$, which influence both treatment and outcome.

Our model, DCT, is designed to learn a mapping from the patient history $\bar{\mathbf{H}}_t$ to these three latent factors $(z_t^{\mathrm{I}}, z_t^{\mathrm{O}}, z_t^{\mathrm{C}})$.

## 4 METHODOLOGY

We propose the **Disentangled Causal Transformer (DCT)**, an encoder-decoder architecture that mitigates confounding bias by disentangling representations into underlying causal components. The encoder-decoder framework is designed for functional specialization: the encoder is dedicated to building a comprehensive representation of the time-series, while the decoder leverages this rich context to perform causal disentanglement. This design enables the model to simultaneously achieve high performance in both temporal modeling and causal debiasing, thereby overcoming the common trade-off between factual and counterfactual prediction accuracy.

### 4.1 DCT ARCHITECTURE OVERVIEW

The architecture of our Disentangled Causal Transformer (DCT) is depicted in Figure 2. It features an encoder-decoder structure. The encoder processes the patient's history $\bar{\mathbf{H}}_t$ to generate a hidden representation, which is then mapped into three disentangled causal factor sequences: instrumental ($z^{\mathrm{I}}$), outcome-specific ($z^{\mathrm{O}}$), and confounder ($z^{\mathrm{C}}$). The decoder then uses these factors, along with future treatments and static features, to predict the counterfactual outcome sequence. At the core of this architecture is our **Disentangled Multi-Head Attention (DMHA)** mechanism, which is detailed in the following section.

### 4.2 DISENTANGLED MULTI-HEAD ATTENTION (DMHA)

Standard Multi-Head Attention (MHA) is designed to allow a model to jointly attend to information from different representation subspaces, but offers no guarantee that these heads learn diverse or specialized functions. To enforce structured specialization, we introduce two key modifications that create our Disentangled Multi-Head Attention (DMHA).

First, inspired by Li et al. (2018; 2021), we employ a **diversity regularizer** to encourage different attention heads to capture distinct patterns. Crucially, we make a deliberate design choice to enforce orthogonality *only* on the final head outputs. Our rationale is twofold: (1) it provides a direct incentive for each head's final contribution to be unique, and (2) it avoids over-constraining the learning process, allowing heads flexibility in how they achieve this diversity. This principle is operationalized via the diversity loss, $\mathcal{L}_{\mathrm{div}}$, which penalizes the cosine similarity between the output representations of any two heads before the final linear projection:

$$\mathcal{L}_{\mathrm{div}} = \sum_{i=1}^{H} \sum_{j=i+1}^{H} \frac{|\langle \mathbf{O}_i, \mathbf{O}_j \rangle_F|}{\|\mathbf{O}_i\|_F \|\mathbf{O}_j\|_F} \tag{2}$$

where $\mathbf{O}_i$ is the output of head $i$ and $H$ is the total number of heads. The effectiveness of this focused, output-level regularization—compared to more complex, multi-stage penalties—was confirmed through a rigorous ablation study (see Appendix A.5), validating our streamlined design.

Second, and the core of the DMHA mechanism, is the partitioning of the $H$ attention heads into three groups: $\mathcal{H}_I$ for instrumental factors, $\mathcal{H}_O$ for outcome-specific factors, and $\mathcal{H}_C$ for confounding factors. For each causal factor $f \in \{I, O, C\}$, its corresponding representation $\mathbf{Z}^f$ is generated by first concatenating the outputs of all heads in its dedicated group $\mathcal{H}_f$, and then projecting the result:

$$\mathbf{Z}^f = \underset{i \in \mathcal{H}_f}{\mathrm{Concat}} \left( \mathrm{head}_i(\mathbf{Q}, \mathbf{K}, \mathbf{V}) \right) \mathbf{W}_O^f \tag{3}$$

where each individual head's output is a attention function with its own projection matrices with $\mathcal{L}_{\mathrm{div}}$ in Eq 2:

$$\mathrm{head}_i(\mathbf{Q}, \mathbf{K}, \mathbf{V}) = \mathrm{Attention}(\mathbf{Q}\mathbf{W}_Q^i, \mathbf{K}\mathbf{W}_K^i, \mathbf{V}\mathbf{W}_V^i) \tag{4}$$

While the diversity loss $\mathcal{L}_{\mathrm{div}}$ encourages diversity among individual heads, it does not explicitly guarantee that the aggregated representations for each causal factor group are distinct. To address this, we introduce an additional regularization term, $\mathcal{L}_{\mathrm{sep}}$, which penalizes the cosine similarity between the final representations of the different factor groups:

$$\mathcal{L}_{\mathrm{sep}} = \sum_{\substack{f_a, f_b \in \{I, O, C\} \\ f_a \neq f_b}} \frac{|\langle \mathbf{Z}^{f_a}, \mathbf{Z}^{f_b} \rangle_F|}{\|\mathbf{Z}^{f_a}\|_F \|\mathbf{Z}^{f_b}\|_F} \tag{5}$$

where $\mathbf{Z}^f$ is the aggregated output representation from Eq. 3.

### 4.3 Encoder-Decoder Structure

**Encoder** The DCT encoder is a stack of $N$ identical blocks that maps the patient's history $\bar{\mathbf{H}}_t$ to three latent causal factor sequences. Each encoder block employs a Two-Stage Attention (TSA) mechanism Zhang & Yan (2023) to efficiently capture both time-wise and channel-wise dependencies. The final layer of the encoder outputs a hidden representation sequence $\mathbf{H}_{\mathrm{enc}}$. This sequence is then projected through **disentangled tower** to produce the disentangled latent factor sequence $z^f$ for each factor $f \in \{I, O, C\}$.

$$\begin{aligned} \mathbf{H}_{\mathrm{enc}} &= \mathrm{Encoder}(\mathrm{Embed}(\bar{\mathbf{H}}_t)) \\ z^f &= \mathrm{Linear}_f(\mathbf{H}_{\mathrm{enc}}) \end{aligned} \tag{6}$$

**Decoder** The decoder consists of $N$ blocks, with the final block being a specialized **Disentangled Causal Transformer Block**. This block fuses causal representation disentanglement with time-series modeling as follows:

Firstly, the standard multi-head self-attention sub-layer is replaced with our DMHA mechanism, which processes the hidden state of the decoder from the previous layer, $\mathbf{H}_{\mathrm{dec}, N-1}$, partitioning it into three specific queries:

$$\mathbf{Z}_{\mathrm{sa}}^I, \mathbf{Z}_{\mathrm{sa}}^O, \mathbf{Z}_{\mathrm{sa}}^C = \mathrm{DMHA}(\mathbf{H}_{\mathrm{dec}, N-1}, \mathbf{H}_{\mathrm{dec}, N-1}, \mathbf{H}_{\mathrm{dec}, N-1}) \tag{7}$$

Secondly, the standard cross-attention sub-layer is replaced by three parallel, structurally constrained MHA modules. Each attention module uses one of the $\mathbf{Z}_{sa}^f$ as its query to exclusively attend to its corresponding latent factors $z^f$ from the encoder. This design strictly prevents information leakage between different causal pathways:

$$\mathbf{Z}_{\mathrm{cross}}^f = \mathrm{MHA}_f \left( \mathbf{Z}_{sa}^f, z^f, z^f \right) \quad \text{for } f \in \{I, O, C\} \tag{8}$$

Finally, each pathway is updated independently via a residual connection, followed by a feed-forward network (FFN). The block's final outputs are three fully updated, disentangled representations, $\mathbf{Z}_{\mathrm{dec}}^f, f \in \{I, O, C\}$:

$$\mathbf{Z}_{\mathrm{dec}}^f = \mathrm{FFN}_f \left( \mathrm{LayerNorm} \left( \mathbf{Z}_{\mathrm{sa}}^f + \mathbf{Z}_{\mathrm{cross}}^f \right) \right) \tag{9}$$

These three output sequences are then channeled into dedicated prediction heads to fulfill their respective causal roles via a multi-task learning objective:

The instrument-specific ($\mathbf{Z}_{\text{dec}}^{\text{I}}$) and confounder ($\mathbf{Z}_{\text{dec}}^{\text{C}}$) representations are used to predict the probability of treatment.

The outcome-specific ($\mathbf{Z}_{\text{dec}}^{\text{O}}$) and confounder ($\mathbf{Z}_{\text{dec}}^{\text{C}}$) representations, along with the treatment $A_t$, are used to predict the outcome.

## 4.4 TRAINING OBJECTIVE

To ensure the disentangled representations learn their intended information, we design a comprehensive multi-task objective function.

**Multi-Task Prediction Losses**   To enforce the specified causal relationships and ensure the correct information flow within our framework (Fig. 1), we introduce two tasks.

- **Outcome Prediction**:   The outcome $Y_{t+1}$ should be predictable from the outcome representations($z_t^{\text{O}}$) and confounders ($z_t^{\text{C}}$) with treatment $A_t$. The outcome prediction loss is a weighted mean squared error:

$$\mathcal{L}_O = \omega_t \cdot \left\| Y_{t+1} - f_O \left( \text{Concat}(z_t^{\text{O}}, z_t^{\text{C}}), A_t \right) \right\|^2 \tag{10}$$

  where $f_O$ is an MLP prediction head and $\omega_t$ are weights learned via Eq. 13.

- **Treatment Prediction**: The treatment $A_t$ should be predictable from the instrument representations ($z_t^{\text{I}}$) and confounders ($z_t^{\text{C}}$). The treatment prediction loss is a cross-entropy loss:

$$\mathcal{L}_T = \text{CrossEntropy}\left( A_t, f_T \left( \text{Concat}(z_t^{\text{I}}, z_t^{\text{C}}) \right) \right) \tag{11}$$

**Causal Regularization Losses**   To achieve the desired disentanglement and mitigate confounding bias, we introduce two distinct regularization losses.

- **Representation Balance Loss**: This loss enforces independence between the outcome-specific representation $O$ and the treatment $A$, a principle motivated by prior work Hassanpour & Greiner (2019b) and illustrated in our framework (Fig. 1). We implement this by using the Maximum Mean Discrepancy (MMD) to minimize the statistical dependence between the distributions of the latent factors $z^{\text{O}}$ across different treatment groups:

$$\mathcal{L}_{\text{mmd}} = \sum_{a \in \mathcal{A}} \text{MMD}\left( \{z_{t,i}^{\text{O}}\}_{i:A_i=a}, \{z_{t,j}^{\text{O}}\}_{j:A_j \neq a} \right) \tag{12}$$

- **Covariate Balancing Loss**: While re-weighting is a common approach for mitigating confounding bias, prior methods often rely on learning propensity scores Hassanpour & Greiner (2019b); Lim (2018). The effectiveness of such methods is highly dependent on the robustness of the propensity prediction task. Inspired by Imai & Ratkovic (2014); Wu et al. (2023), Instead of relying on unstable IPTW Imai & Ratkovic (2014), we learn sample weights $\omega_i$ that directly balance the distribution of the confounding factor $z^{\text{C}}$ across different treatment groups. The weights are learned by minimizing:

$$\mathcal{L}_{\text{balance}} = \sum_{a \in \mathcal{A}} \text{MMD}\left( \{\omega_{t,i} z_{t,i}^{\text{C}}\}_{i:A_i=a}, \{\omega_{t,j} z_{t,j}^{\text{C}}\}_{j:A_j \neq a} \right) \tag{13}$$

**Final Objective Function**   The total loss for DCT is a weighted sum of all components:

$$\mathcal{L}_{\text{total}} = \lambda_O \mathcal{L}_O + \lambda_T \mathcal{L}_T + \lambda_{\text{mmd}} \mathcal{L}_{\text{mmd}} + \lambda_{\text{balance}} \mathcal{L}_{\text{balance}} + \lambda_{\text{dis}} \mathcal{L}_{\text{dis}} \tag{14}$$

where $\mathcal{L}_{\text{dis}}$ comprises the regularizers defined in Section 4.2, and the hyperparameters $\lambda_{(\cdot)}$ are used to weigh the contribution of each term.

## 5 EXPERIMENTS

We conduct a comprehensive set of experiments on both fully-synthetic and semi-synthetic datasets to evaluate the performance of the Disentangled Causal Transformer (DCT). Detailed hyperparameter configurations, the weights of loss items and implementation specifics are provided in A.1.

Table 1: Results for $\tau$-step-ahead prediction on the synthetic benchmark. Performance is evaluated under varying levels of time-varying confounding ($\gamma$). Results are averaged over five runs (lower is better, best in bold).

|  |  | $\tau = 2$ | $\tau = 3$ | $\tau = 4$ | $\tau = 5$ | $\tau = 6$ |
|---|---|---|---|---|---|---|
| $\gamma = 0$ | RMSNs | $0.74 \pm 0.04$ | $0.78 \pm 0.05$ | $0.82 \pm 0.07$ | $0.85 \pm 0.09$ | $0.89 \pm 0.10$ |
|  | CRN | $\mathbf{0.66 \pm 0.05}$ | $0.69 \pm 0.05$ | $0.72 \pm 0.05$ | $0.76 \pm 0.05$ | $0.80 \pm 0.05$ |
|  | CT | $0.68 \pm 0.06$ | $0.70 \pm 0.05$ | $0.73 \pm 0.05$ | $0.76 \pm 0.05$ | $0.80 \pm 0.05$ |
|  | DCT (Ours) | $\mathbf{0.66 \pm 0.05}$ | $\mathbf{0.68 \pm 0.05}$ | $\mathbf{0.69 \pm 0.05}$ | $\mathbf{0.71 \pm 0.05}$ | $\mathbf{0.73 \pm 0.06}$ |
| $\gamma = 1$ | RMSNs | $0.79 \pm 0.07$ | $0.81 \pm 0.06$ | $0.86 \pm 0.08$ | $0.91 \pm 0.09$ | $0.95 \pm 0.11$ |
|  | CRN | $0.67 \pm 0.05$ | $0.69 \pm 0.04$ | $0.72 \pm 0.03$ | $0.76 \pm 0.03$ | $0.79 \pm 0.03$ |
|  | CT | $0.67 \pm 0.04$ | $0.70 \pm 0.04$ | $0.74 \pm 0.04$ | $0.78 \pm 0.04$ | $0.81 \pm 0.04$ |
|  | DCT (Ours) | $\mathbf{0.65 \pm 0.03}$ | $\mathbf{0.67 \pm 0.03}$ | $\mathbf{0.69 \pm 0.02}$ | $\mathbf{0.71 \pm 0.02}$ | $\mathbf{0.74 \pm 0.02}$ |
| $\gamma = 2$ | RMSNs | $0.79 \pm 0.05$ | $0.85 \pm 0.05$ | $0.93 \pm 0.10$ | $1.01 \pm 0.15$ | $1.08 \pm 0.19$ |
|  | CRN | $0.74 \pm 0.04$ | $0.82 \pm 0.05$ | $0.90 \pm 0.06$ | $0.98 \pm 0.07$ | $1.05 \pm 0.08$ |
|  | CT | $0.74 \pm 0.07$ | $0.79 \pm 0.08$ | $0.85 \pm 0.09$ | $0.89 \pm 0.11$ | $0.93 \pm 0.11$ |
|  | DCT (Ours) | $\mathbf{0.67 \pm 0.03}$ | $\mathbf{0.70 \pm 0.02}$ | $\mathbf{0.74 \pm 0.02}$ | $\mathbf{0.77 \pm 0.02}$ | $\mathbf{0.80 \pm 0.02}$ |
| $\gamma = 3$ | RMSNs | $0.94 \pm 0.11$ | $1.06 \pm 0.20$ | $1.20 \pm 0.23$ | $1.33 \pm 0.29$ | $1.44 \pm 0.36$ |
|  | CRN | $0.94 \pm 0.14$ | $1.16 \pm 0.26$ | $1.35 \pm 0.38$ | $1.51 \pm 0.47$ | $1.64 \pm 0.54$ |
|  | CT | $0.90 \pm 0.11$ | $0.98 \pm 0.13$ | $1.05 \pm 0.14$ | $1.11 \pm 0.14$ | $1.16 \pm 0.14$ |
|  | DCT (Ours) | $\mathbf{0.75 \pm 0.08}$ | $\mathbf{0.84 \pm 0.10}$ | $\mathbf{0.88 \pm 0.11}$ | $\mathbf{0.91 \pm 0.12}$ | $\mathbf{0.96 \pm 0.12}$ |
| $\gamma = 4$ | RMSNs | $1.28 \pm 0.29$ | $1.47 \pm 0.41$ | $1.56 \pm 0.43$ | $1.60 \pm 0.42$ | $1.61 \pm 0.37$ |
|  | CRN | $1.15 \pm 0.15$ | $1.37 \pm 0.22$ | $1.58 \pm 0.29$ | $1.76 \pm 0.34$ | $1.89 \pm 0.37$ |
|  | CT | $1.31 \pm 0.52$ | $1.51 \pm 0.59$ | $1.68 \pm 0.67$ | $1.81 \pm 0.70$ | $1.89 \pm 0.70$ |
|  | DCT (Ours) | $\mathbf{1.02 \pm 0.21}$ | $\mathbf{1.14 \pm 0.23}$ | $\mathbf{1.22 \pm 0.24}$ | $\mathbf{1.30 \pm 0.20}$ | $\mathbf{1.35 \pm 0.23}$ |

## 5.1 EXPERIMENTAL SETUP

**Datasets and baselines** Model performance is assessed on two widely used standard benchmarksBica et al. (2020); Vaswani et al. (2017); Melnychuk et al. (2022): a fully-synthetic dataset and a semi-synthetic dataset derived from a real-world clinical database.

**Fully-Synthetic Dataset:** The fully-synthetic benchmark provides a controlled environment where we can precisely vary the strength of time-varying confounding via a parameter, $\gamma$. Higher values of $\gamma$ indicate stronger confounding bias, allowing for a targeted evaluation of the model's robustness.

**Semi-Synthetic Dataset:** The semi-Synthetic benchmark aims to bridge the gap to real-world complexity where ground-truth counterfactuals are unobservable, leverage real patient trajectories from the MIMIC-III clinical database Johnson et al. (2016) for their realistic covariate structures. Combine the high-dimensional complexity of real clinical data with a known causal ground truth for rigorous evaluation.

We compare DCT against several state-of-the-art models for longitudinal counterfactual prediction: RMSN Lim (2018),CRN Bica et al. (2020), and Causal Transformer (CT) Melnychuk et al. (2022), details of benchmarks and baselines are in A.2 A.4.

**Evaluation Metrics.** We use the Root Mean Squared Error (RMSE) over the prediction horizon $\tau$ as our primary evaluation metric. To ensure reliable and statistically stable comparisons, all reported metrics are averaged over five independent runs.

## 5.2 RESULTS ON FULLY-SYNTHETIC DATA

The results on the fully-synthetic benchmark, presented in Table 1, underscore the superior performance and robustness of DCT, particularly under increasing levels of time-varying confounding.

Across all settings, DCT consistently outperforms the baselines. This advantage becomes especially pronounced as the confounding strength intensifies. For instance, at high confounding levels such as

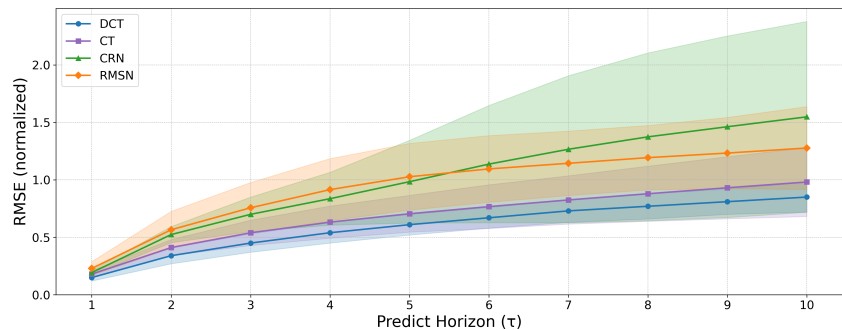

Figure 3: Results on semi-synthetic data for $\tau$-step-ahead prediction. The dataset is based on MIMIC-III, featuring real-world covariate distributions. Results are averaged over five runs (lower is better).

$\gamma = 3$ and $\gamma = 4$, the performance of all baseline models deteriorates sharply, with their prediction errors escalating and variance increasing. In stark contrast, DCT's performance degrades much more gracefully, maintaining a substantial performance margin over the next-best model and exhibiting lower variance.

We attribute this remarkable robustness to DCT's principled architectural design. By explicitly disentangling the latent representation into instrumental, outcome, and confounding factors, DCT can surgically target and mitigate confounding bias without corrupting outcome relevant information. Baseline models, which apply indiscriminate balancing to the entire representation, are caught in a difficult trade-off: to control for strong confounding, they risk over-adjustment by discarding valuable outcome-predictive signals that are correlated with the treatment. DCT's causal disentanglement mechanism sidesteps this pitfall entirely, enabling both effective bias control and robust information preservation. This ensures stable and accurate counterfactual predictions even under severe confounding pressure.

Table 2: Ablation Study of DCT Component Contributions to Performance (Lower is Better)

| Model Variation | $\tau = 2$ | $\tau = 4$ | $\tau = 6$ | $\tau = 8$ | $\tau = 10$ |
|---|---|---|---|---|---|
| DCT (Full Model) | **0.34** | **0.54** | **0.67** | **0.77** | **0.85** |
| w/o Disentanglement | 0.37 | 0.60 | 0.76 | 0.90 | 0.99 |
| w/o MMD Regularization | 0.36 | 0.57 | 0.72 | 0.85 | 0.94 |
| w/o Balance Regularization | 0.36 | 0.56 | 0.73 | 0.84 | 0.95 |

## 5.3 RESULTS ON SEMI-SYNTHETIC DATA

As illustrated in Figure 3, DCT robustly outperforms all baseline methods on the semi-synthetic benchmark across the entire prediction horizon ($\tau = 1$ to $10$). Crucially, this performance gap is not static; it progressively widens as the prediction horizon extends, highlighting DCT's superior long-term stability.

This trend is particularly telling, as long-term forecasting is precisely where models are most vulnerable to both the decay of historical information and the amplification of confounding bias. We attribute DCT's sustained superiority to the powerful synergy between its two core components. Its Transformer backbone provides the inherent capacity to model complex, long-range temporal dependencies, while the causal disentanglement mechanism ensures that this process is not corrupted by compounding bias. By simultaneously addressing both temporal modeling and causal inference challenges, DCT maintains its predictive accuracy over long horizons where other methods falter.

## 5.4 ABLATION STUDY

To validate the contribution of each component in our proposed model, we conducted a rigorous ablation study. We remove parts of the Disentangled Causal Transformer (DCT) to quantify the impact of its core mechanisms on counterfactual prediction. The results are summarized in Table 2.

Table 3: Sensitivity analysis of auxiliary loss weights ($\lambda$) on the semi-synthetic dataset. We report performance (RMSE) at selected time horizons.

| Loss Component | Weight Multiplier | $\tau$=2 | $\tau$=4 | $\tau$=6 | $\tau$=8 | $\tau$=10 |
|---|---|---|---|---|---|---|
| **Base (Ours)** | $1\times$ | **0.34** | **0.54** | **0.67** | **0.77** | **0.85** |
| $\mathcal{L}_T$ (BCE) | $5\times$ | 0.34 | 0.55 | 0.69 | 0.80 | 0.89 |
| | $10\times$ | 0.34 | 0.56 | 0.71 | 0.83 | 0.90 |
| $\mathcal{L}_{\mathrm{mmd}}$ | $5\times$ | 0.34 | 0.54 | 0.67 | 0.77 | 0.85 |
| | $10\times$ | 0.34 | 0.55 | 0.69 | 0.79 | 0.87 |
| $\mathcal{L}_{\mathrm{balance}}$ | $5\times$ | 0.34 | 0.54 | 0.67 | 0.78 | 0.85 |
| | $10\times$ | 0.34 | 0.55 | 0.68 | 0.79 | 0.87 |
| $\mathcal{L}_{\mathrm{dis}}$ | $5\times$ | 0.35 | 0.56 | 0.71 | 0.84 | 0.91 |
| | $10\times$ | 0.35 | 0.56 | 0.71 | 0.84 | 0.91 |

As hypothesized, the full DCT model consistently outperforms all ablated versions across all time horizons. The most critical component is evidently the **representation disentanglement** achieved by our DMHA mechanism. Removing this mechanism (**w/o disentangle**)—which effectively reverts DCT to a standard attention-based encoder-decoder architecture—causes a dramatic performance drop. This culminates in a 16.5% relative increase in error at $\tau = 10$ (from 0.85 to 0.99), demonstrating that our architectural approach to disentanglement is the primary driver of DCT's superior performance on longitudinal data.

The causal regularization losses are also integral to DCT's success. Removing the outcome representation balance loss (**w/o mmd**) or the confounder balancing loss (**w/o balance**) results in a consistent performance decline, leading to an increase in relative error 10.6% and 11.8% at $\tau = 10$, respectively. This confirms that each of the balance losses works as designed.

We further analyze the model's sensitivity to the weights ($\lambda$) of its auxiliary losses to validate the robustness of our multi-task objective. We scaled each weight individually by factors of $5\times$ and $10\times$ relative to our base configuration. The results, shown in Table 3, indicate that our model is highly robust to these variations. For instance, increasing the weight of $\mathcal{L}_{\mathrm{mmd}}$ or $\mathcal{L}_{\mathrm{balance}}$ by $10\times$ results in only a minor increase in long-term prediction error (e.g., RMSE at $\tau = 10$ increases from 0.85 to 0.87). While our default weights consistently yield the best performance across all horizons, the minimal performance decay under significant weight changes underscores that our model's effectiveness is not contingent on precise hyperparameter tuning.

In summary, these results collectively validate our design choices, demonstrating that each component of DCT — from its core disentanglement architecture to its specific causal regularizers — contributes meaningfully to achieving state-of-the-art performance.

## 6 CONCLUSION

In this paper, we address a fundamental challenge in longitudinal causal inference: the trade-off between factual and counterfactual prediction accuracy, which arises from the prevailing paradigm of indiscriminate covariate balancing. We introduced the Disentangled Causal Transformer (DCT), the first architecture to integrate causal representation disentanglement within the powerful Transformer framework. This design enables confounding bias adjustment while preserving the full signal for outcome prediction, effectively unifying factual and counterfactual prediction accuracy, achieving state-of-the-art performance in counterfactual prediction. This work lays the foundation for several future directions. An immediate avenue is to explore the modularity of our Disentangled Multi-Head Attention (DMHA). It could potentially serve as a drop-in replacement for standard attention in encoder-only or decoder-only Transformers, and investigating the performance implications of such an adaptation is a promising research question. Broader future work includes addressing key limitations, such as unobserved confounding and representation identifiability, and extending the DCT framework to other causal tasks in real-world clinical decision-support systems.

## 7 REPRODUCIBILITY STATEMENT

We are committed to ensuring the full reproducibility of the results presented in this paper. To facilitate this, we provide the following resources:

**Experimental Setup.** Comprehensive details of our experimental setup are documented in Appendix A.1. This includes all model hyperparameters, training configurations, and the optimization settings used for our experiments.

**Datasets.** A thorough description of all datasets used in this study is available in Appendix A.2. This includes details on their sources, preprocessing steps, and the specific data splits for training, validation, and testing.

We believe that these measures provide sufficient detail for other researchers to replicate our findings and build upon our work.

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

## A  APPENDIX

### A.1  IMPLEMENTATION DETAILS

All experiments were conducted on a single NVIDIA Tesla V100 GPU using the PyTorch framework. Consistent with prior work Lim (2018); Bica et al. (2020); Melnychuk et al. (2022), we train our model via a two-stage procedure, applying teacher forcing and utilizing the Adam optimizer Kingma (2014). and disabled during evaluation. This two-stage approach aligns with our encoder-decoder architecture, enforcing a functional specialization: the encoder is dedicated to building a comprehensive representation of the time-series, while the decoder leverages this rich context to perform causal disentanglement. To enhance training stability and improve generalization, we apply an Exponential Moving Average (EMA) to the model's parameters. Detailed hyperparameter configurations are provided in Table 4. On the fully-synthetic benchmark, we adopted distinct hyperparameter sets for varying levels of confounding strength, a practice consistent with prior work Melnychuk et al. (2022); Bica et al. (2020). To ensure a fair comparison, all baseline models were configured by strictly replicating the hyperparameters reported in Melnychuk et al. (2022).

The decoder consists of (i) standard vanilla Transformer blocks and (ii) a Disentangled Causal Transformer block. This specialized DCT block integrates (a) standard Multi-Head Attention (MHA) and (b) our proposed Disentangled Multi-Head Attention (DMHA). DMHA partitions its attention heads into three functionally distinct groups to model instrumental variables, confounders, and target factors, respectively. To maintain expressive capacity for each of the factors, the number of heads in DMHA is tripled, which correspondingly triples the block's hidden dimension.

The overall training objective for DCT, $\mathcal{L}_{\text{total}}$, is a weighted sum of the primary outcome prediction loss ($\mathcal{L}_O$) and several auxiliary regularization terms:

$$\mathcal{L}_{\text{total}} = \lambda_O \mathcal{L}_O + \lambda_T \mathcal{L}_T + \lambda_{\text{mmd}} \mathcal{L}_{\text{mmd}} + \lambda_{\text{balance}} \mathcal{L}_{\text{balance}} + \lambda_{\text{dis}} \mathcal{L}_{\text{dis}} \tag{15}$$

To prioritize the main prediction task, we set $\lambda_O = 1.0$, while the weights for all auxiliary objectives are uniformly set to 0.1.

Table 4: Hyperparameter configurations for the DCT model on the fully-synthetic and semi-synthetic benchmarks.

| Model | Module | Hyperparameter | Fully-Synthetic | Semi-Synthetic |
|---|---|---|---|---|
| DCT | Encoder | Transformer Blocks | 2 | 2 |
| | | Learning Rate | {0.01, 0.001} | 0.01 |
| | | Batch Size | 512 | 128 |
| | | Dropout Rate | 0.1 | 0.2 |
| | | Attention Heads (MHA) | 2 | 3 |
| | | Hidden Dimension | 18 | 42 |
| | | Training Epochs | 200 | 400 |
| | Decoder | Transformer Blocks | 2 | 1 |
| | | Disentangled Transformer Blocks | 1 | 1 |
| | | Learning Rate | {0.01, 0.001} | 0.001 |
| | | Batch Size | 1024 | 512 |
| | | Attention Heads (MHA) | 2 | 2 |
| | | Attention Heads (DMHA) | 6 | 6 |
| | | Hidden Dimension (MHA) | 18 | 42 |
| | | Hidden Dimension (DMHA) | 54 | 126 |
| | | Training Epochs | 150 | 200 |

### A.2  DETAILS OF BENCHMARK DATASETS

We evaluate our proposed model, DCT, on two benchmarks widely adopted in previous works Melnychuk et al. (2022); Bica et al. (2020). The first is a fully-synthetic benchmark Geng et al. (2017), enabling a controlled assessment of confounding. The second is a semi-synthetic benchmark derived

from real-world clinical data Johnson et al. (2016), designed to assess model robustness in a setting that more closely mirrors clinical reality.

### A.2.1 FULLY-SYNTHETIC BENCHMARK

We use the Tumor Growth (TG) simulator Geng et al. (2017) as our primary synthetic Data Generating Process (DGP), configuring it to yield a one-dimensional outcome ($\mathbf{Y}_t \in \mathbb{R}$, $d_y = 1$).

**Outcome Simulation** The simulation involves two binary treatments, chemotherapy ($A_t^{(c)}$) and radiotherapy ($A_t^{(r)}$), with distinct temporal effects. Radiotherapy has an immediate effect $d_t$, while chemotherapy's influence, $C(t)$, is prolonged and decaying. The tumor volume $\mathbf{Y}_t$ evolves according to a multiplicative update rule that integrates natural growth with treatment effects:

$$\mathbf{Y}_{t+1} = \left(1 + \rho \log\left(\frac{K}{\mathbf{Y}_t}\right) - \beta_c C_t - (\alpha_r d_t + \beta_r d_t^2)\right)\mathbf{Y}_t + \epsilon_t \mathbf{Y}_t, \tag{16}$$

where $\rho, K, \beta_c, \alpha_r, \beta_r$ are simulation constants, and $\epsilon_t \sim \mathcal{N}(0, 0.01^2)$ is independently sampled noise. To model patient heterogeneity, the treatment response parameters ($\beta_c, \alpha_r, \beta_r$) are sampled from a mixture of truncated normal distributions.

**Time-Varying Confounding** Time-varying confounding is introduced by conditioning treatment assignments on past outcomes. Both treatments, $A_t^c$ and $A_t^r$, are assigned via the stochastic policy:

$$A_t^c, A_t^r \sim \text{Bernoulli}\left(\sigma\left(\frac{\gamma}{D_{\max}}\left(D_{15}(\bar{\mathbf{Y}}_{t-1}) - D_{\max}/2\right)\right)\right), \tag{17}$$

where the assignment probability is a sigmoid function of the average tumor diameter over the preceding 15 days, $D_{15}(\bar{\mathbf{Y}}_{t-1})$. The hyperparameter $\gamma$ controls confounding strength: $\gamma = 0$ recovers a randomized trial, while larger values induce stronger confounding.

**Evaluation and Dataset Generation** Consistent with prior work Bica et al. (2020); Melnychuk et al. (2022), we generate datasets for each confounding level $\gamma$, comprising 10,000 trajectories for training, 1,000 for validation, and 1,000 for testing, with a maximum length of 60 time steps. We evaluate models via counterfactual prediction. For one-step-ahead evaluation, all $2^2 = 4$ potential outcomes are generated. For multi-step-ahead evaluation (horizon $\tau_{\max}$), we assess policies involving a single treatment applied at various future times, yielding $2(\tau_{\max} - 1)$ distinct counterfactual trajectories.

### A.2.2 SEMI-SYNTHETIC BENCHMARK

We construct a semi-synthetic benchmark derived from the MIMIC-III dataset Johnson et al. (2016), adopting the data generation protocol used in Melnychuk et al. (2022). First, we select a cohort of 1000 patients with ICU stays between 20 and 100 hours. This cohort is divided into training (60%), validation (20%), and testing (20%) sets. For each patient, we synthesize a complete data trajectory via a four-step process.

**Step 1: Generating Latent Untreated Trajectories** The latent untreated trajectory $Z_t^{j,(i)}$ for each patient $i$ and outcome $j$ is generated as the sum of endogenous, exogenous, and noise components:

$$Z_t^{j,(i)} = \underbrace{\alpha_s^j \text{B-spline}(t) + \alpha_g^j g^{j,(i)}(t)}_{\text{endogenous}} + \underbrace{\alpha_f^j f_Z^{j,(i)}(\mathbf{X}_t^j)}_{\text{exogenous}} + \underbrace{\epsilon_t^{j,(i)}}_{\text{noise}}, \tag{18}$$

where the noise term is $\epsilon_t^{j,(i)} \sim \mathcal{N}(0, 0.005^2)$, and $\alpha_s^j$, $\alpha_g^j$, and $\alpha_f^j$ are weight parameters. The endogenous term combines a global trend with local, patient-specific variations. The global trend, represented by B-spline(t), is sampled from a mixture of three cubic splines (modeling rapid decline, mild decline, and stable trajectories). The patient-specific variations, $g(\cdot)$, are drawn from a Gaussian Process (GP). The exogenous term $f_Z(\cdot)$ models covariate dependencies and is approximated using Random Fourier Features (RFF).

**Step 2: Simulating a Treatment Plan**  We assign $d_a$ binary treatments $A_t^l \sim \text{Bernoulli}(p_t^l)$, where the assignment probability $p_t^l$ depends on past outcomes $\bar{\mathbf{Y}}_{t-1}$ and current covariates $\mathbf{X}_t$:

$$p_t^l = \sigma(\gamma_A^l \bar{A}_{t_l}(\bar{\mathbf{Y}}_{t-1}) + \gamma_X^l f_Y^l(\mathbf{X}_t) + b_l). \tag{19}$$

Here, $\gamma_A^l$ and $\gamma_X^l$ control the confounding strength, and the function $f_Y^l(\cdot)$ is another GP approximated via RFF.

**Step 3: Simulating Treatment Effects**  The treatment effect $E^j(t)$ is an additive term that aggregates the influence of past treatments within a time window $t - w^l, \ldots, t$:

$$E^j(t) = \sum_{i=t-w^l}^{t} \frac{\min_{l=1,\ldots,d_a}(\mathbb{I}[A_i^l = 1] \cdot p_i^l \cdot \beta_{l,j})}{(w^l - i)^2}, \tag{20}$$

where $\beta_{l,j}$ is the maximal effect of treatment $l$ on outcome $j$.

**Step 4: Generating Observed Outcomes**  The final observed outcome $Y_t^j$ is the sum of the untreated latent trajectory and the cumulative treatment effect:

$$Y_t^j = Z_t^j + E^j(t). \tag{21}$$

**Experimental Setup and Evaluation**  In our experiments, we set the number of synthetic treatments to $d_a = 3$ and outcomes to $d_y = 2$. For one-step-ahead evaluation, we generate all $2^3 = 8$ potential outcomes. For multi-step-ahead evaluation (with horizon $\tau_{\max} = 10$), we sample 10 random counterfactual trajectories for each patient at each time step.

### A.3 Assumptions for Causal Identification

Our objective is to estimate counterfactual outcomes under time-varying interventions from observational data. The ability to identify these causal quantities—that is, to express them purely in terms of the distribution of observed data—hinges on three foundational assumptions, adapted from the potential outcomes framework Rubin (1978) for sequential settings Robins & Hernan (2008).

- **Consistency** The potential outcome under the received treatment sequence coincides with the observed outcome. *Formally, if $\bar{\mathbf{A}}_t = \bar{\mathbf{a}}_t$, then $\mathbf{Y}_{t+1}[\bar{\mathbf{a}}_t] = \mathbf{Y}_{t+1}$.*

- **Sequential Overlap** At any time, for any given patient history, there is a non-zero probability of receiving any possible treatment. *Formally, for any history $\bar{\mathbf{h}}_t$ with $\mathbb{P}(\bar{\mathbf{H}}_t = \bar{\mathbf{h}}_t) > 0$, we require $0 < \mathbb{P}(A_t = a_t \mid \bar{\mathbf{H}}_t = \bar{\mathbf{h}}_t) < 1$.*

- **Sequential Ignorability (No Unobserved Confounding)** Conditional on the observed history, the current treatment assignment is independent of the potential outcomes. This implies that the history $\bar{\mathbf{H}}_t$ captures all confounders. *Formally, $A_t \perp\!\!\!\perp \mathbf{Y}_{t+1}[\mathbf{a}_t] \mid \bar{\mathbf{H}}_t$, for all possible treatments $\mathbf{a}_t$.*

### A.4 Baselines

- **RMSN:** The Recurrent Marginal Structural Network (RMSN) Lim (2018) is a seminal re-weighting method that operationalizes Marginal Structural Models (MSMs) with Recurrent Neural Networks. It employs a multi-task architecture where one RNN-based component estimates time-varying treatment probabilities to compute Inverse Probability of Treatment Weights (IPTWs). A second component then uses these weights to train a sequence-to-sequence outcome model on a re-weighted, pseudo-randomized population, thereby adjusting for time-varying confounding.

- **CRN:** The Counterfactual Recurrent Network (CRN) Bica et al. (2020) is a foundational representation-learning approach designed to mitigate confounding bias. It uses an RNN encoder and employs domain-adversarial training to achieve balance. Specifically, an adversary network is trained to predict the treatment assignment from the learned patient representations. The encoder, in turn, is trained to generate representations that are indistinguishable (invariant) to this adversary, thus enforcing that the representation distribution is similar across treatment arms. Potential outcomes are subsequently predicted from these balanced representations.

- **CT:** The Causal Transformer (CT) Melnychuk et al. (2022) advances the representation-learning paradigm by replacing the RNN encoder with a more powerful Transformer architecture. This architectural shift is motivated by the Transformer's superior ability to capture complex and long-range dependencies within patient trajectories. In addition to leveraging adversarial training similar to CRN, CT introduces a novel balancing regularizer, the Covariate Deconfounding Condition (CDC) loss. This loss directly minimizes a distance metric (e.g., MMD) between the representation distributions of the treated and control groups, offering a more direct mechanism for achieving covariate balance.

## A.5 DETAILS ON AUXILIARY REGULARIZATION TERMS

In the design of our Disentangled Multi-Head Attention (DMHA), we initially explored a composite penalty term to promote orthogonality between the representations learned by each head at three key stages. Our initial hypothesis was that enforcing diversity at multiple levels of the attention mechanism would be most effective. While our final streamlined design, informed by rigorous validation, retains only the output orthogonality term (see Section 4.2), we document the auxiliary regularization terms explored during our investigation here for completeness. This documentation also serves as a useful "negative result" that may guide future research in this area.

The two auxiliary components we tested were:

**Subspace Orthogonality.** To ensure heads draw upon different feature subspaces, we penalized the cosine similarity between their projected value matrices.

$$\mathcal{L}_{\text{value}} = \sum_{i=1}^{H} \sum_{j=i+1}^{H} \frac{|\langle \mathbf{V}_i, \mathbf{V}_j \rangle_F|}{\|\mathbf{V}_i\|_F \|\mathbf{V}_j\|_F} \tag{22}$$

where $\mathbf{V}_i = \mathbf{V}\mathbf{W}_V^i$ is the value matrix for head $i$.

**Attention Disagreement.** To compel heads to focus on different input positions, we directly penalized the overlap between their attention weight matrices.

$$\mathcal{L}_{\text{attn}} = \sum_{i=1}^{H} \sum_{j=i+1}^{H} \langle \mathbf{A}_i, \mathbf{A}_j \rangle_F \tag{23}$$

where $\mathbf{A}_i$ is the attention matrix (after softmax) for head $i$.

The ablation study in Table 5 yielded a valuable insight: while enforcing orthogonality on the final head outputs ($\mathcal{L}_{\text{div}}$ in the main text) is critical, a more granular regularization on intermediate stages via $\mathcal{L}_{\text{value}}$ and $\mathcal{L}_{\text{attn}}$ slightly hindered performance. This empirical finding demonstrates that our final design is not arbitrary but the result of rigorous validation.

Table 5: Impact of Regularization Terms in Disentangled Multi-Head Attention (DMHA) on Performance (Lower is Better).

| # | Regularization | | | $\tau = 2$ | $\tau = 10$ |
|---|---|---|---|---|---|
| | *Out.* | *Attn.* | *Sub.* | | |
| 1 | × | × | × | 0.35 | 0.88 |
| 2 | ✓ | × | × | **0.34** | **0.85** |
| 3 | ✓ | ✓ | × | 0.36 | 0.89 |
| 4 | ✓ | ✓ | ✓ | 0.36 | 0.95 |

## A.6 EMPIRICAL VERIFICATION OF SEMANTIC DISENTANGLEMENT

A critical consideration for our model is the challenge of disentangling causal factors—Instrumental (I), Outcome (O), and Confounder (C)—given that perfect separation is a theoretical ideal rarely achieved in practice due to signal overlap. A core design principle of DCT is that it does not assume this split *a priori*; instead, its architecture is designed to *learn* a meaningful semantic separation in the latent space. To empirically validate this learned separation, we conducted a series of probing experiments to test whether the disentangled representations adhere to their intended causal roles.

### A.6.1 Probing for Treatment Information in the Outcome Representation

This experiment tests whether the outcome representation $z_O$ has been successfully purged of treatment-predictive information. We introduced a new prediction head (a two-layer MLP) tasked with predicting the treatment $A$ using *only* the learned $z_O$. For comparison, the baseline performance is derived from the model's intended treatment prediction pathway, which uses both $z_I$ and $z_C$.

Table 6: Probing for Treatment ($A$) leakage in $z_O$ on the Semi-Synthetic dataset.

| Representation Used | Performance (AUC) |
|---|---|
| $z_I, z_C$ (Base) | 0.74 |
| $z_O$ only (Probing Task) | **0.51 (Random Guessing)** |

As shown in Tables 6, attempting to predict treatment from $z_O$ consistently yields an AUC statistically equivalent to random guessing ($\approx 0.5$).

### A.6.2 Probing for Outcome Information in the Instrumental Representation

This second experiment tests whether the instrumental representation $z_I$ has been purged of outcome-predictive information. We added a new prediction head (a two-layer MLP) tasked with predicting outcome to use *only* $z_I$ and evaluated its performance against the base model.

Table 7: Probing for Outcome ($Y$) leakage in $z_I$ on the Semi-Synthetic dataset.

| Representation Used | $\tau$=1 | $\tau$=3 | $\tau$=5 | $\tau$=7 | $\tau$=9 | $\tau$=10 |
|---|---|---|---|---|---|---|
| $z_O, z_C$ (Base) | 0.15 | 0.45 | 0.61 | 0.73 | 0.81 | 0.85 |
| $z_I$ only (Probing Task) | 1.23 | 1.22 | 1.22 | 1.24 | 1.25 | 1.24 |

The results in Table 7 show a complete performance collapse when predicting from $z_I$. The RMSE is orders of magnitude higher than the base model, confirming that the instrumental representation contains negligible outcome-relevant information.

### A.6.3 Conclusion of Probing Experiments

In summary, these probing experiments provide powerful, direct empirical evidence for our model's core mechanism. They confirm that the separation learned by DCT is a **meaningful semantic disentanglement**, not an arbitrary partitioning, and that the architecture successfully routes information into the correct causal pathways. While perfect disentanglement remains a theoretical ideal, these results demonstrate that our architecture effectively learns to approximate this separation—a capability far beyond what is achievable with indiscriminate balancing.

### A.7 Robustness to Stronger Confounding

To assess the scalability and robustness of our approach under more challenging conditions, we evaluated DCT against baselines on the fully-synthetic dataset with significantly higher confounding strengths ($\gamma = 6, 8, 10$). The results are presented in Table 8. While baseline models exhibit drastic performance degradation under these severe confounding scenarios, DCT maintains a significant performance advantage and superior stability. For instance, at $\tau = 10$ and $\gamma = 6$, DCT's RMSE (1.467) is substantially lower than that of CT (2.456) and CRN (3.471). This validates the effectiveness of our disentanglement approach in extreme settings.

### A.8 Sensitivity to MMD Kernel Choice

To investigate the model's sensitivity to the choice of discrepancy metric, we conducted an ablation study comparing the default Gaussian kernel with a common alternative, the Inverse Multi-Quadric (IMQ) kernel, for the $\mathcal{L}_{\mathrm{mmd}}$ loss. As shown in Table 9, the performance is highly robust to this

Table 8: Performance on the Fully-Synthetic Dataset with Higher Confounding Strength ($\gamma$). RMSE is reported (lower is better).

| $\gamma$ | Method | $\tau = 2$ | $\tau = 4$ | $\tau = 6$ | $\tau = 8$ | $\tau$=10 |
|---|---|---|---|---|---|---|
| | RMSN | 2.429 | 2.592 | 2.562 | 2.435 | 2.302 |
| $\gamma = 6$ | CRN | 1.991 | 2.716 | 3.104 | 3.352 | 3.471 |
| | CT | 2.156 | 2.369 | 2.462 | 2.498 | 2.456 |
| | **DCT (Ours)** | **1.341** | **1.461** | **1.518** | **1.514** | **1.467** |
| | RMSN | 5.782 | 6.093 | 8.782 | 11.854 | 14.019 |
| $\gamma = 8$ | CRN | 3.966 | 4.337 | 4.462 | 4.527 | 4.467 |
| | CT | 5.782 | 6.093 | 8.782 | 11.854 | 14.019 |
| | **DCT (Ours)** | **3.778** | **4.018** | **4.047** | **3.943** | **3.731** |
| | RMSN | 5.148 | 5.763 | 6.058 | 6.220 | 6.230 |
| $\gamma = 10$ | CRN | 7.426 | 9.083 | 9.324 | 9.200 | 8.905 |
| | CT | 5.148 | 5.763 | 6.058 | 6.220 | 6.230 |
| | **DCT (Ours)** | **4.976** | **5.487** | **5.421** | **5.245** | **4.898** |

choice. The RMSE for the IMQ kernel (0.83 at $\tau = 10$) is only marginally different from the Gaussian kernel (0.85). This reinforces that DCT's strong performance stems from its core architectural design rather than a specific, fine-tuned configuration of its auxiliary losses.

Table 9: Ablation Study on MMD Kernel Choice on the Semi-Synthetic Dataset.

| Loss | Kernel | $\tau$=1 | $\tau$=2 | $\tau$=3 | $\tau$=4 | $\tau$=5 | $\tau$=6 | $\tau$=7 | $\tau$=8 | $\tau$=9 | $\tau$=10 |
|---|---|---|---|---|---|---|---|---|---|---|---|
| $\mathcal{L}_{\text{mmd}}$ | Gaussian (Base) | 0.15 | 0.34 | 0.45 | 0.54 | 0.61 | 0.67 | 0.73 | 0.77 | 0.81 | 0.85 |
| | IMQ | 0.15 | 0.35 | 0.46 | 0.53 | 0.60 | 0.68 | 0.74 | 0.78 | 0.82 | 0.83 |

## A.9 ON THE NECESSITY OF $\mathcal{L}_{\text{SEP}}$

Our DMHA mechanism employs a two-level disentanglement strategy. To empirically ablate the contribution of the second level—which enforces separation between the final aggregated representations of causal factors—we evaluated a variant of DCT without the $\mathcal{L}_{\text{sep}}$ loss (Eq. 5). The results in Table 10 show a consistent and increasing trend of performance degradation when $\mathcal{L}_{\text{sep}}$ is removed. The performance gap widens as $\tau$ increases, reaching a +5.9% relative error at $\tau = 10$. This demonstrates that $\mathcal{L}_{\text{sep}}$ is a crucial component for ensuring the long-term stability and accuracy of our counterfactual predictions.

Table 10: Ablation study on the effect of $\mathcal{L}_{\text{sep}}$. Relative RMSE increase is shown in parentheses.

| Model | $\tau$=2 | $\tau$=4 | $\tau$=6 | $\tau$=8 | $\tau$=10 |
|---|---|---|---|---|---|
| Base (Full Model) | 0.34 | 0.54 | 0.67 | 0.77 | 0.85 |
| w/o $\mathcal{L}_{\text{sep}}$ | 0.34 (0.0%) | 0.54 (0.0%) | 0.68 (+1.5%) | 0.79 (+2.6%) | 0.90 (+5.9%) |

## A.10 RUNTIME ANALYSIS AND COMPUTATIONAL COMPLEXITY

To provide a clear picture of the computational requirements of DCT, we report its runtime performance in comparison to baselines. The training and inference times on the semi-synthetic benchmark, measured on a single NVIDIA V100 GPU, are detailed in Table 11.

**Justification of the Trade-off.** The results show that DCT's increased computational cost corresponds directly to substantial performance improvements, especially over longer prediction horizons (see Figure 3). This highlights that the architectural sophistication is a deliberate design choice,

Table 11: Runtime analysis on the Semi-Synthetic dataset (1 NVIDIA V100 GPU).

| Model | Training Time (min) | Inference Time (min) |
|---|---|---|
| RMSN | 42 | 1 |
| CRN | 44 | 1.5 |
| CT | 130 | 3 |
| DCT (Ours) | 161 | 5 |

representing a trade-off between computational resources and predictive fidelity. In high-stakes domains such as personalized medicine, where accuracy is paramount, we believe that this trade-off is not only justified but essential.

### A.11 USE OF LARGE LANGUAGE MODELS (LLMS)

In preparing this manuscript, LLMs was employed for the sole purpose of language polishing. Its function was strictly confined to refining grammar, enhancing clarity, and improving the style of the text. No part of the research ideation, methodology, data analysis, or conclusions was generated or influenced by the LLM.

