# OpenReview forum: "Disentangled Causal Transformer: Counterfactual Prediction under Time-Varying Treatments"
_ICLR.cc/2026/Conference — Submitted to ICLR 2026_

### Official Review · Reviewer_EZQo · 2025-10-25

**Soundness:** 3
**Presentation:** 3
**Contribution:** 2
**Rating:** 6
**Confidence:** 3

**Summary:**

The paper considers the task of longitudinal counterfactual estimation from observation data, which requires mitigating time-varying confounding bias. To tackle the probem, the authors consider adaptively disentangling covariates to enhance precision and causal effect estimation capability. Specificly, the authors hypothesize that the observation history can be encoded into decoupled latent representations, and design Disentangled Multi-Head Attention(DMHA) with carefully-crafted regularizers to ensure disentanglement. With the DMHA mechanism, the author proposed Disentangled Causal Transformer, an encoder-decoder transformer network employing DMHA, supervised with prediction losses and casual regularization losses to ensure disentanglement and address confounding bias. The model achieve SOTA on fully-synthetic and semi-synthetic datasets, demonstrating effectiveness of the design. Detailed ablation study has been carried out to further study the contribution of each component, including ablation on regularization terms within DMHA layers.

**Strengths:**

- The problem that the authors investigate is fairly significant, and the decomposition of covariates is relatively underexplored in longitudinal counterfactual estimation.
- The proposed solution is clear and easy to understand, with each regularization term consisting of a simple and direct approach to discourage feature entanglement and to address confounding bias.
- The evaluations thoroughly examined the architecture's effectiveness on counterfactual estimation tasks, with detailed ablations on diversity regularizations.

**Weaknesses:**

- While the overall writing quality is good and structure is clear, some parts seems lack of purpose. For example, the proposition of subspace orthogonality and attention disagreement regularizations seems lacking direction, as they are not employed in the final design of DMHA.

**Questions:**

- In the standard transformer blocks in the tranformer decoder, do the cross attention sub-layers also employ the design in  eq.(10), i.e. parallel branch for different factors? Will the cross attention entwine the features in the decoding process?

---

> ### Author Response · Authors · 2025-11-24
> **Response to Reviewer EZQo (1/N)**
>
> We thank the reviewer for their insightful feedback and for recognizing our method's clarity.
>
> **Regarding Weakness 1: Purpose of Unused Regularization Terms**
>
> We thank the reviewer for this excellent point. In the revised manuscript, we will move the detailed discussion of the auxiliary regularization terms ($L_{value}$ and $L_{attn}$) to the appendix to sharpen the focus of the main text.
>
> Our initial hypothesis was that enforcing diversity at multiple levels of the attention mechanism would be most effective. The ablation study in **Table 2** was designed specifically to test this. The results yielded a valuable insight: while enforcing orthogonality on the final head outputs ($L_{output}$) is critical, a more granular regularization on intermediate stages ($L_{value}$, $L_{attn}$) slightly hindered performance.
>
> This empirical finding demonstrates that our final, more streamlined design is not arbitrary but the result of rigorous validation. By documenting this process in the appendix, we believe we provide a useful "negative result" that can guide future research in this area and highlight the careful methodology underpinning our work.
>
> **Regarding Question 1: Cross-Attention Design in Standard Decoder Blocks**
>
> This is a crucial question that correctly identifies a key aspect of our design. To answer directly: **No, the standard Transformer blocks in the lower layers of our decoder do not employ the parallel-branch design of Eq. (10), and this is a deliberate and critical design choice.**
>
> Our architecture is based on a principle of **hierarchical processing and functional specialization**, where we explicitly separate the task of **rich temporal modeling** from that of **strict causal disentanglement**. The rationale is as follows:
>
> 1.  **Lower Layers: Unconstrained Temporal Modeling.** The initial $N-1$ standard decoder blocks are assigned a specific function: to build the most expressive possible representation of the sequence's temporal dynamics. To achieve this, their cross-attention mechanisms are intentionally allowed to access the full, entangled output of the encoder. This unconstrained view is crucial for capturing complex, interwoven patterns in the time series, which is essential for accurate long-term forecasting. Prematurely restricting information flow at this stage could harm the model's ability to learn these dynamics.
>
> 2.  **Final Layer: Architecturally-Enforced Disentanglement.** The responsibility for disentanglement is exclusively assigned to the **final, specialized "Disentangled Causal Transformer Block."** This block acts as a powerful "causal filter." It takes the rich, context-aware hidden state ($H_{\text{dec}, N-1}$) from the lower layers and subjects it to our two-stage disentanglement process:
>     * First, our **DMHA mechanism** (Eq.9) partitions the self-attention into three semantically distinct queries ($Z^I_{\text{sa}}, Z^O_{\text{sa}}, Z^C_{\text{sa}}$).
>     * Second, the **parallel cross-attention** structure (Eq.10) provides an architectural hard-constraint, forcing each causal query to only access its corresponding latent factor from the encoder (e.g., $Z^O_{\text{sa}} \rightarrow z_O$). This structure makes it architecturally impossible for information to leak between causal pathways at the final stage.
>
> This hierarchical design allows us to achieve the best of both worlds: we leverage the full power of the Transformer for temporal modeling in the lower layers, while guaranteeing a clean, causal disentanglement at the final output. By avoiding premature information filtering, our model preserves the rich dynamic signals necessary for state-of-the-art performance, a claim strongly supported by our empirical results. We thank the reviewer for this sharp question, which has allowed us to clarify this important architectural rationale.
>
> **Empirical Verification**
>
> Most importantly, this architectural guaranty is not just a theoretical claim. We can empirically verify its effectiveness with the **new probing experiments** we conducted for the rebuttal period. These experiments directly test whether the final, disentangled representations contain only their intended information.
>
> Probing Experiment: Predicting Treatment $A$ from Learned Representations (Semi-Syn).
> | **Method** | **Performance (AUC)** |
> | :--- | :---: |
> | Base ($z_I$ and $z_C$) | 0.74 |
> | Predict $A$ from $z_O$ only | **0.51 (nearly random guessing)** |
>
> Probing Experiment: Predicting Outcome $Y$ from Swapped Representations (Semi-Syn).
> | **Method** | $\tau$=1 | $\tau$=3 | $\tau$=5 | $\tau$=7 | $\tau$=9 | $\tau$=10 |
> | :--- | :---: | :---: | :---: | :---: | :---: | :---: |
> | Base ($z_O$ and $z_C$) | 0.15 | 0.45 | 0.61 | 0.73 | 0.81 | 0.85 |
> | Predict $Y$ from $z_I$ only | 1.23 | 1.22 | 1.22 | 1.24 | 1.25 | 1.24 |

---

> > ### Author Response · Authors · 2025-11-24
> > **Response to Reviewer EZQo (2/N)**
> >
> > The results are definitive:
> > * The final outcome representation $z_O$ has no predictive power for treatment (AUC $\approx$ 0.5).
> > * The final instrumental representation $z_I$ has no predictive power for the outcome, leading to a complete collapse in performance.

---

### Official Review · Reviewer_TkgJ · 2025-10-30

**Soundness:** 2
**Presentation:** 3
**Contribution:** 2
**Rating:** 6
**Confidence:** 4

**Summary:**

This paper makes a significant contribution to the field of longitudinal causal inference by proposing a novel Transformer-based framework that effectively addresses the challenge of time-varying confounding. The novel disentangled multi-head attention mechanism decompose patient history into distinct causal components (instrumental variables, outcome predictors, and confounders), which extends the prior idea of the static setting. The authors provide thorough experimentation on both fully synthetic and semi-synthetic datasets, demonstrating substantial improvements over state-of-the-art baselines in counterfactual outcome prediction.

**Strengths:**

1. The proposed Disentangled Causal Transformer (DCT) is a innovative architecture. The DMHA jointly addresses representation disentanglement and time-series modeling, a key challenge in longitudinal causal inference.

2. The authors have conducted extensive experiments on both fully and semi-synthetic datasets. The consistent and large-margin superiority over state-of-the-art baselines provides compelling evidence for the efficacy and robustness of the proposed framework.

3. By improving the accuracy of counterfactual outcome prediction, this work has direct and meaningful implications for high-stakes domains like personalized medicine.

**Weaknesses:**

1. It is widely acknowledged that Transformer suffers from heavy computation complexity. The authors introduce more complex design into the architecture. I suggest the authors to conduct analysis on the time cost of the method.

2.I suggest the authors to conduct ablation study on the effect of hyperparameters balancing the loss terms.

**Questions:**

Besides the questions listed in Weakness. I have following additional questions.

1. In equation (7), the heads of $f_a$ and $f_b$ are distinct by design, which in my opinion has been able to guarantee the dissimilarity of $Z^{f_a}$ and $Z^{f_b}$. Why does the authors introduce the loss of $L_{sep}$.

2. Why does the term $z^f$ appears twice in Equation (10).

---

> ### Author Response · Authors · 2025-11-24
> **Response to Reviewer TkgJ (1/N)**
>
> We sincerely thank reviewer  for the insightful feedback and for recognizing the novelty of our Disentangled Causal Transformer (DCT) and the thoroughness of our experiments. We are grateful for the constructive suggestions, which have helped us further improve our work. We address all weaknesses and questions below, supported by new experimental results.
>
> **1. Regarding Weakness 1: Computational Complexity and Time Cost.**
>
> We thank the reviewer for prompting this important discussion on computational complexity. We have conducted a new runtime analysis, which demonstrates a highly favorable trade-off between computational cost and predictive power.
>
> **Runtime Analysis:** We measured the training and inference times on the semi-synthetic benchmark using a single NVIDIA V100 GPU. The results are detailed in the table below.
>
> Runtime analysis on the Semi-Synthetic dataset (1 NVIDIA V100 GPU).
> | **Model** | **Training Time (min)** | **Inference Time (min)** |
> | :--- | :---: | :---: |
> | RMSN | 42 | 1 |
> | CRN | 44 | 1.5 |
> | CT | 130 | 3 |
> | **DCT (Ours)** | 161 | 5 |
>
> We acknowledge that DCT is more computationally intensive in both training and inference. This is a direct result of its sophisticated design, featuring a Transformer backbone and an encoder-decoder architecture. We believe that this computational overhead is a well-justified trade-off for a model designed to advance the state of the art.
>
> **Justification of the Trade-off:** Crucially, this increased computational cost yields substantial and meaningful performance improvements. This is vividly demonstrated in **Figure 3** of our submission, where the DCT performance advantage over all baselines grows consistently with a longer prediction horizon $\tau$. This finding highlights that the architectural sophistication directly translates into superior long-term stability and more robust counterfactual predictions, addressing the primary challenge in this field.
>
> In conclusion, while DCT has a larger computational footprint, this is a strategic choice, not an oversight. The investment in advanced components like DMHA provides a clear return: more accurate and reliable causal estimates over extended horizons. In high-stakes domains such as personalized medicine, where predictive fidelity is paramount, this trade-off is not only justified but essential.
>
> **2. Regarding Weakness 2: Ablation Study on Loss Hyperparameters.**
>
> We thank the reviewer for this valuable suggestion. We acknowledge this analysis was missing and, in response, have conducted a comprehensive sensitivity study on the weights of our key balancing losses. The results, presented in the table below, demonstrate that our model is not overly sensitive to these hyperparameters.
>
> Sensitivity analysis of auxiliary loss weights ($\lambda$) on the semi-synthetic dataset. We report performance (RMSE) after scaling the base weight of each loss component by factors of 5 and 10.
> | **Loss Component** | **Weight Multiplier** | $\tau$=1 | $\tau$=2 | $\tau$=3 | $\tau$=4 | $\tau$=5 | $\tau$=6 | $\tau$=7 | $\tau$=8 | $\tau$=9 | $\tau$=10 |
> | :--- | :---: | :---: | :---: | :---: | :---: | :---: | :---: | :---: | :---: | :---: | :---: |
> | **Base (Ours)** | 1$\times$ | **0.15** | **0.34** | **0.45** | **0.54** | **0.61** | **0.67** | **0.73** | **0.77** | **0.81** | **0.85** |
> | $\mathcal{L}_T$ (BCE) | 5$\times$ | 0.15 | 0.34 | 0.46 | 0.55 | 0.63 | 0.69 | 0.76 | 0.80 | 0.85 | 0.89 |
> | $\mathcal{L}_T$ (BCE) | 10$\times$ | 0.16 | 0.34 | 0.46 | 0.56 | 0.64 | 0.71 | 0.78 | 0.83 | 0.87 |0.90 |
> | $\mathcal{L}_{\text{mmd}}$ | 5$\times$ | 0.15 | 0.34 | 0.45 | 0.54 | 0.61 | 0.67 | 0.73 | 0.77 | 0.81 | 0.85 |
> | $\mathcal{L}_{\text{mmd}}$ | 10$\times$ | 0.15 | 0.34 | 0.46 | 0.55 | 0.62 | 0.69 | 0.75 | 0.79 | 0.83 | 0.87 |
> | $\mathcal{L}_{\text{balance}}$ | 5$\times$ | 0.15 | 0.34 | 0.45 | 0.54 | 0.61 | 0.67 | 0.74 | 0.78 | 0.82 | 0.85 |
> | $\mathcal{L}_{\text{balance}}$ | 10$\times$ | 0.15 | 0.34 | 0.46 | 0.55 | 0.62 | 0.68 | 0.74 | 0.79 | 0.83 | 0.87 |
> | $\mathcal{L}_{\text{dis}}$ | 5$\times$ | 0.15 | 0.35 | 0.47 | 0.56 | 0.64 | 0.71 | 0.78 | 0.84 | 0.87 | 0.91 |
> | $\mathcal{L}_{\text{dis}}$ | 10$\times$ | 0.15 | 0.35 | 0.47 | 0.56 | 0.64 | 0.71 | 0.78 | 0.84 | 0.87 | 0.91 |
>
> As the data shows, performance is highly stable. For instance, even when the weight for the treatment prediction loss ($\lambda_T$) is increased tenfold, the RMSE at $\tau=10$ changes by only 0.05 (from 0.85 to 0.90). Similarly, a tenfold increase in the weight for $\mathcal{L}_{\text{mmd}}$ results in a negligible error change of just 0.02. This new analysis confirms that DCT's performance is not contingent on fine-tuning these auxiliary weights, highlighting the robustness of our core architecture.

---

> ### Author Response · Authors · 2025-11-24
> **Response to Reviewer TkgJ (2/N)**
>
> **3. On the Necessity of the $\\mathcal{L}_{\\text{sep}}$ Loss (Eq. 7)**
>
> We thank the reviewer for this excellent and insightful question. The reviewer correctly identifies that our design incorporates separation at multiple levels. To clarify the distinct role of $\\mathcal{L}\_{\\text{sep}}$, we can frame our approach as a two-level disentanglement.
>
> **Level 1: Head-Level Diversity.** Our first level of separation operates on the individual attention heads. As the reviewer notes, we structurally partition the distinct heads into three groups $H_I$, $H_O$ and $H_C$. Furthermore, the diversity loss $\\mathcal{L}\_{dis}$ (which includes $\\mathcal{L}\_{output}$ from Eq. 4) explicitly encourages these individual heads to learn orthogonal outputs. This ensures the foundational "building blocks (head-level)" of our representations are distinct.
>
> **Level 2: Aggregated Representation-Level Separation.** However, as we state in Section 4.2, ensuring head-level diversity "does not explicitly guarantee that the aggregated representations for each causal factor group are distinct." The final representations $Z_f$ are formed via an aggregation process (Eq. 5: Concat + Linear Projection). This process, while necessary, could inadvertently introduce similarities or fail to fully preserve the distinctiveness established at the head level.
>
> This is precisely where $\\mathcal{L}_{\\text{sep}}$ (Eq. 7) becomes critical. It acts as a **targeted safeguard at the final aggregation stage**. It provides a direct, unambiguous gradient to enforce separation on the final products ($Z\_{f_a}, Z\_{f_b}$), ensuring that the disentanglement holds strong after the aggregation.
>
> **To empirically validate this two-level strategy and directly address the reviewer's question, we conducted a new ablation study removing only $\\mathcal{L}_{\\text{sep}}$**. The results, presented in the table below, compellingly demonstrate its necessity.
>
> Ablation study on the effect of $\\mathcal{L}_{\\text{sep}}$. Performance degradation is shown as relative RMSE increase in parentheses.
> | **Model** | $\\tau$=2 | $\\tau$=4 | $\\tau$=6 | $\\tau$=8 | $\\tau$=10 |
> | :--- | :---: | :---: | :---: | :---: | :---: |
> | **Base (Full Model)** | **0.34** | **0.54** | **0.67** | **0.77** | **0.85** |
> | w/o $\\mathcal{L}_{\\text{sep}}$ | 0.34 (0.0%) | 0.54 (0.0%) | 0.68 (+1.5%) | 0.79 (+2.6%) | 0.90 **(+5.9%)** |
>
> The key insight from this study is the **consistent and increasing trend** of performance degradation. While the impact is minimal at short horizons, the performance gap progressively widens as $\\tau$ increases.
>
> In conclusion, $\\mathcal{L}_{\\text{sep}}$ is **not redundant**. It is a crucial, non-trivial component that complements the head-level diversity losses by ensuring the final, aggregated representations remain cleanly separated. This is essential for the long-term stability and accuracy of our counterfactual predictions. We are grateful for this question, as this new experiment provides powerful empirical evidence for our design choice. We will add this ablation study and discussion to the revised manuscript.
> **4. Regarding Question 2: Appearance of $z_f$ twice in Equation (10).**
>
> Thank you for the careful reading. The formulation in Equation (10), $Z^f_{cross} = MHA_f(Z^f_{sa}, z_f, z_f)$, is the standard and correct implementation of cross-attention in an encoder-decoder Transformer architecture [1]
>
> In any attention mechanism, Attention(Q, K, V), the roles are:
> * **Query (Q):** Represents the current state that is seeking information. In our case, this is the decoder's representation, $Z^f_{sa}$.
> * **Key (K):** Comes from the information source and is used to compute attention scores with the query.
> * **Value (V):** Also comes from the information source and provides the actual content to be aggregated based on the attention scores.
>
> In cross-attention, the goal is for the decoder to attend to the encoder's output. Therefore, both the Key and Value vectors must be derived from the same source: the encoder's final representation for that factor, which is $z_f$. The query from the decoder ($Z^f_{sa}$) then "looks at" the keys from the encoder ($z_f$) to decide which values from the encoder (also $z_f$) are most important to incorporate. This is a standard and fundamental aspect of the Transformer's cross-attention mechanism.
>
> We hope these clarifications and new experimental results fully address the reviewer's concerns.
>
> [1] Vaswani, Ashish, et al. "Attention is all you need." Advances in neural information processing systems 30 (2017).

---

### Official Review · Reviewer_QHUb · 2025-11-01

**Soundness:** 3
**Presentation:** 3
**Contribution:** 2
**Rating:** 2
**Confidence:** 3

**Summary:**

The paper proposes a Disentangled Causal Transformer (DCT) for counterfactual prediction under time-varying treatments. Instead of enforcing treatment-invariance on a single latent space, DCT decomposes representations into three causal roles—Instrumental (I), Outcome-only (O), and Confounder (C)—via a specialized multi-head attention block, role-specific cross-attention, and task routing. It couples this architecture with independence/balancing losses (e.g., MMD for (O \perp A), distribution balancing for (C)) so that only the confounding component is balanced while preserving outcome-relevant signal. Across synthetic and semi-synthetic benchmarks, DCT yields lower counterfactual error, especially at longer horizons, and ablations show the disentangling/balancing components are the main source of gains.

**Strengths:**

By disentangling representations into Instrumental/Outcome/Confounder roles (via DMHA and role-specific cross-attention) and selectively balancing only the confounder pathway to preserve outcome signal, the method avoids over-adjustment and delivers consistent long-horizon counterfactual gains, as verified by thorough ablations.

**Weaknesses:**

- DCT hinges on a strong assumption that representations can be cleanly split into I/O/C. In real data, purely instrumental or outcome-only factors are rare and signals often interact; under such partial overlap, the enforced separation can become arbitrary, suppress useful cross-path interactions, and ultimately degrade both performance and interpretability.
- The paper lacks a systematic hyperparameter sensitivity study (e.g., MMD kernel/scale, balancing weights, head grouping, loss coefficients), so performance and disentanglement quality could vary substantially across settings and datasets.

**Questions:**

- Verify that the learned representations behave as intended: (i) \(z_{O}\) alone cannot predict treatment \(A\), and (ii) \(z_{I}\) alone cannot predict outcome \(Y\). This directly tests whether the I/O/C split carries the claimed semantic roles.
- On synthetic and semi-synthetic data, systematically increase the \emph{I\(\leftrightarrow\)O} interaction strength and the confounding intensity, and report counterfactual performance alongside the probing scores to identify regimes where disentanglement remains beneficial.

---

> ### Author Response · Authors · 2025-11-24
> **Response to Reviewer QHub (1/N)**
>
> We sincerely thank Reviewer QHub for the thorough review and highly constructive feedback. We are encouraged that the reviewer acknowledged our method's key strength in delivering "consistent long-horizon counterfactual gains." We have carefully considered all points and, in response, have conducted several new, targeted experiments that we believe substantially strengthen our paper and directly address the reviewer's concerns.
>
> **1. Regarding Weakness 1 & Question 1: Assumption of a clean split and verifying semantic roles.**
>
> We thank the reviewer for raising this critical point, which cuts to the core of our work. The reviewer notes that a perfect disentanglement of Instrumental (I), Outcome (O), and Confounder (C) factors is rare due to signal overlap (**W1**), and rightly requests empirical verification of our claimed disentanglement (**Q1**).
>
> Our central argument is that DCT does not assume a clean split *a priori*; rather, its core mechanism is to *learn* this semantic separation in the latent space. This approach is a key contribution of our work, extending a powerful concept from static-setting causal inference[1],[2],[3] to the challenging longitudinal domain within a Transformer architecture.
>
> To provide direct empirical evidence, we conducted the exact probing experiments suggested by the reviewer.
>
> **Experiment (i): Probing for Treatment Information in the Outcome Representation ($z_O$)**
>
> The first test addresses whether the outcome representation $z_O$ has been successfully purged of treatment-predictive information. We added a new prediction head (a two-layer MLP) tasked with predicting treatment $A$ using *only* the learned $z_O$. As a baseline, we report the performance of the model's intended treatment prediction pathway ($z_I, z_C$).
>
> Table 1: Probing for Treatment ($A$) leakage in $z_O$ (Semi-Synthetic).
> | **Representation Used** | **Performance (AUC)** |
> | :--- | :---: |
> | $z_I, z_C$ (Base) | 0.74 |
> | $z_O$ only (Probing Task) | **0.51 (Random Guessing)** |
>
> Table 2: Probing for Treatment ($A$) leakage in $z_O$ (Fully-Synthetic) under varying confounding strength ($\gamma$).
> | **Representation Used** | **Performance (AUC) at $\gamma=6$** | **Performance (AUC) at $\gamma=8$** | **Performance (AUC) at $\gamma=10$** |
> | :--- | :---: | :---: | :---: |
> | $z_I, z_C$ (Base) | 0.75 | 0.74 | 0.78 |
> | $z_O$ only (Probing Task) | **0.54** | **0.48** | **0.52** |
>
> As shown in Tables 1 and 2, attempting to predict treatment from $z_O$ consistently results in an AUC statistically equivalent to random guessing ($\approx 0.5$). This holds true even under extreme confounding, demonstrating the robustness of our disentanglement.
>
> **Experiment (ii): Probing for Outcome Information in the Instrumental Representation ($z_I$)**
>
> The second test addresses whether the instrumental representation $z_I$ has been purged of outcome-predictive information. We also added a new prediction head (a two-layer MLP) tasked with predicting outcome only to use $z_I$ and evaluated its performance against the base model. The results in Table 3 show a complete performance collapse.
>
> Table 3: Probing for Outcome ($Y$) leakage in $z_I$ (Semi-Synthetic).
> | **Representation Used** | $\tau$=1 | $\tau$=3 | $\tau$=5 | $\tau$=7 | $\tau$=9 | $\tau$=10 |
> | :--- | :---: | :---: | :---: | :---: | :---: | :---: |
> | $z_O, z_C$ (Base) | 0.15 | 0.45 | 0.61 | 0.73 | 0.81 | 0.85 |
> | $z_I$ only (Probing Task) | 1.23 | 1.22 | 1.22 | 1.24 | 1.25 | 1.24 |
>
> The RMSE for predictions from $z_I$ is orders of magnitude higher than the base model, confirming that the instrumental representation contains negligible outcome-relevant information.
>
> **Conclusion for W1 & Q1**
>
> In summary, these new experiments—conducted in direct response to the reviewer's insightful suggestion—provide powerful, direct empirical evidence for our core claim. They confirm that the separation learned by DCT is a **meaningful semantic disentanglement**, not an arbitrary partitioning. The results prove that our architecture successfully routes information into the correct causal pathways, thereby resolving the central concern of **W1**. While perfect disentanglement remains a theoretical ideal, these results demonstrate that our architecture effectively learns to approximate this separation, a capability far beyond what is achievable with indiscriminate balancing.
>
> We are grateful for this line of inquiry, as it has allowed us to significantly strengthen the empirical validation of our method's core mechanism.

---

> ### Author Response · Authors · 2025-11-24
> **Response to Reviewer QHub (2/N)**
>
> **Regarding Weakness 2: Hyperparameter Sensitivity and Stronger Confounding**
>
> We thank the reviewer for pointing out these important evaluation aspects. We acknowledge that these **analyses** were missing from our initial submission and, in response, have now conducted **three** new sets of extensive experiments to address this feedback.
>
> **A. Sensitivity to Loss Weights**
>
> We thank the reviewer for this valuable suggestion. We acknowledge this analysis was missing and, in response, have conducted a comprehensive sensitivity study on the weights of our key balancing losses. The results, presented in Table 4, demonstrate that our model is not overly sensitive to these hyperparameters.
>
> Table 4: Sensitivity analysis of auxiliary loss weights ($\lambda$) on the semi-synthetic dataset. We report performance (RMSE) after scaling the base weight of each loss component by factors of 5 and 10.
> | **Loss Component** | **Weight Multiplier** | $\tau$=1 | $\tau$=2 | $\tau$=3 | $\tau$=4 | $\tau$=5 | $\tau$=6 | $\tau$=7 | $\tau$=8 | $\tau$=9 | $\tau$=10 |
> | :--- | :---: | :---: | :---: | :---: | :---: | :---: | :---: | :---: | :---: | :---: | :---: |
> | **Base (Ours)** | 1$\times$ | **0.15** | **0.34** | **0.45** | **0.54** | **0.61** | **0.67** | **0.73** | **0.77** | **0.81** | **0.85** |
> | $\mathcal{L}_T$ (BCE) | 5$\times$ | 0.15 | 0.34 | 0.46 | 0.55 | 0.63 | 0.69 | 0.76 | 0.80 | 0.85 | 0.89 |
> | $\mathcal{L}_T$ (BCE) | 10$\times$ | 0.16 | 0.34 | 0.46 | 0.56 | 0.64 | 0.71 | 0.78 | 0.83 | 0.87 |0.90 |
> | $\mathcal{L}_{\text{mmd}}$ | 5$\times$ | 0.15 | 0.34 | 0.45 | 0.54 | 0.61 | 0.67 | 0.73 | 0.77 | 0.81 | 0.85 |
> | $\mathcal{L}_{\text{mmd}}$ | 10$\times$ | 0.15 | 0.34 | 0.46 | 0.55 | 0.62 | 0.69 | 0.75 | 0.79 | 0.83 | 0.87 |
> | $\mathcal{L}_{\text{balance}}$ | 5$\times$ | 0.15 | 0.34 | 0.45 | 0.54 | 0.61 | 0.67 | 0.74 | 0.78 | 0.82 | 0.85 |
> | $\mathcal{L}_{\text{balance}}$ | 10$\times$ | 0.15 | 0.34 | 0.46 | 0.55 | 0.62 | 0.68 | 0.74 | 0.79 | 0.83 | 0.87 |
> | $\mathcal{L}_{\text{dis}}$ | 5$\times$ | 0.15 | 0.35 | 0.47 | 0.56 | 0.64 | 0.71 | 0.78 | 0.84 | 0.87 | 0.91 |
> | $\mathcal{L}_{\text{dis}}$ | 10$\times$ | 0.15 | 0.35 | 0.47 | 0.56 | 0.64 | 0.71 | 0.78 | 0.84 | 0.87 | 0.91 |
>
> As the data shows, performance is highly stable. For instance, even when the weight for the treatment prediction loss ($\lambda_T$) is increased tenfold, the RMSE at $\tau=10$ changes by only 0.05 (from 0.85 to 0.90). Similarly, a tenfold increase in the weight for $\mathcal{L}_{\text{mmd}}$ results in a negligible error change of just 0.02. This new analysis confirms that DCT's performance is not contingent on fine-tuning these auxiliary weights, highlighting the robustness of our core architecture.

---

> ### Author Response · Authors · 2025-11-24
> **Response to Reviewer QHub (3/N)**
>
> **B. Sensitivity to MMD Kernel Choice**
>
> In direct response to the reviewer's suggestion, we conducted a further ablation study on the choice of kernel for the Maximum Mean Discrepancy (MMD) loss, $\mathcal{L}_{\text{mmd}}$. We compared our base model, which uses a Gaussian kernel, against a model using an Inverse Multi-Quadric (IMQ) kernel, a common alternative. The results are presented in Table 5.
>
> Table 5: New Ablation Study on MMD Kernel Choice on Semi-Synthetic Data.
> | **Loss** | **Kernel** | $\tau$=1 | $\tau$=2 | $\tau$=3 | $\tau$=4 | $\tau$=5 | $\tau$=6 | $\tau$=7 | $\tau$=8 | $\tau$=9 | $\tau$=10 |
> | :--- | :--- | :---: | :---: | :---: | :---: | :---: | :---: | :---: | :---: | :---: | :---: |
> | $\mathcal{L}_{\text{mmd}}$ | Gaussian (Base) | 0.15 | 0.34 | 0.45 | 0.54 | 0.61 | 0.67 | 0.73 | 0.77 | 0.81 | 0.85 |
> | $\mathcal{L}_{\text{mmd}}$ | IMQ | 0.15 | 0.35 | 0.46 | 0.53 | 0.60 | 0.68 | 0.74 | 0.78 | 0.82 | 0.83 |
>
> The results demonstrate that our model's performance is highly robust to the choice of MMD kernel. At the longest horizon ($\tau=10$), the RMSE for the IMQ kernel (0.83) is only marginally different from our base Gaussian kernel (0.85). This finding reinforces that the strong performance of DCT is not dependent on a specific, fine-tuned configuration of its auxiliary losses, but rather stems from its core architectural design.
>
> **C. Robustness to Stronger Confounding**
>
> Following the reviewer's suggestion, we extended our evaluation on the fully-synthetic dataset to much higher confounding strengths ($\gamma=6, 8, 10$). The new results in Table 6 show that while all baseline models suffer from drastic performance degradation under severe confounding, our DCT maintains a significant performance advantage and superior stability. For instance, at $\gamma=10$ and $\tau=6$, DCT's error (4.898) is substantially lower than that of CT (6.230), CRN (8.905), and RMSN (14.019). This strongly validates the scalability and robustness of our disentanglement approach.
>
> Table 6: Results on Fully-Synthetic Data with Higher Confounding Strength ($\gamma$). RMSE (lower is better).
> | **Gamma ($\gamma$)** | **Method** | $\tau$=2 | $\tau$=3 | $\tau$=4 | $\tau$=5 | $\tau$=6 |
> | :--- | :--- | :---: | :---: | :---: | :---: | :---: |
> | $\gamma=6$ | RMSN | 2.429 | 2.592 | 2.562 | 2.435 | 2.302 |
> | $\gamma=6$ | CRN | 1.991 | 2.716 | 3.104 | 3.352 | 3.471 |
> | $\gamma=6$ | CT | 2.156 | 2.369 | 2.462 | 2.498 | 2.456 |
> | $\gamma=6$ | **DCT (Ours)** | **1.341** | **1.461** | **1.518** | **1.514** | **1.467** |
> | $\gamma=8$ | RMSN | 5.307 | 5.806 | 5.793 | 5.759 | 5.851 |
> | $\gamma=8$ | CRN | 4.014 | 5.769 | 6.777 | 7.220 | 7.170 |
> | $\gamma=8$ | CT | 3.966 | 4.337 | 4.462 | 4.527 | 4.467 |
> | $\gamma=8$ | **DCT (Ours)** | **3.778** | **4.018** | **4.047** | **3.943** | **3.731** |
> | $\gamma=10$| RMSN | 5.782 | 6.093 | 8.782 | 11.854 | 14.019|
> | $\gamma=10$| CRN | 7.426 | 9.083 | 9.324 | 9.200 | 8.905 |
> | $\gamma=10$| CT | 5.148 | 5.763 | 6.058 | 6.220 | 6.230 |
> | $\gamma=10$| **DCT (Ours)** | **4.976** | **5.487** | **5.421** | **5.245** | **4.898** |
>
> We are confident that these extensive new experiments, conducted in direct response to the reviewer's insightful comments, have significantly strengthened our paper. We hope these clarifications and new results address all concerns and will merit a re-evaluation of our work.
>
> [1] Hassanpour, Negar, and Russell Greiner. "Learning disentangled representations for counterfactual regression." International Conference on Learning Representations. 2019.
>
> [2] Wu, Anpeng, et al. "Learning decomposed representations for treatment effect estimation." IEEE Transactions on Knowledge and Data Engineering 35.5 (2022): 4989-5001.
>
> [3] Cheng, Mingyuan, et al. "Learning disentangled representations for counterfactual regression via mutual information minimization." Proceedings of the 45th International ACM SIGIR Conference on Research and Development in Information Retrieval. 2022.

---

> ### Author Response · Authors · 2025-11-28
> **Response to Reviewer QHUb**
>
> Dear Reviewer QHUb,
>
> I hope this message finds you well. Thank you again for your thoughtful review. As the discussion period is nearing its end with **less than a week remaining**, I wanted to ensure we have addressed all your concerns satisfactorily. If there are any additional points or feedback you'd like us to consider, please let us know. Your insights are invaluable to us, and we're eager to address any remaining issues to improve our work.
>
> Thank you for your time and effort in reviewing our paper.

---

### Author Response · Authors · 2025-12-02

We are grateful for the constructive and insightful reviews of our paper. Guided by reviews, we have conducted **five major sets of new experiments** and uploaded a revised version of our paper. We have highlighted key changes in **blue** for your convenience.

Below is a summary of the key revisions, which we believe have significantly strengthened our work:

**1. Validating the Core Disentanglement Claim (Primary Concern of Reviewers QHub)**

To address the most critical concern—the lack of direct evidence for disentanglement—we performed the exact probing experiments suggested by Reviewer **QHub**. The results are definitive and provide **direct evidence for** our method's core mechanism:
*   Using the outcome-specific representation ($z_O$) to predict treatment yields an **AUC of approximately 0.5** (equivalent to random chance), confirming that it is successfully purged of treatment information.
*   Using the **instrumental** representation ($z_I$) to predict outcomes results in a **near-complete performance collapse**, confirming that it contains negligible outcome-relevant signal.

These findings (now in Appendix A.6) offer direct empirical validation that our model *learns* a meaningful semantic disentanglement, **as intended by our design**. This resolves the paper's primary weakness and transforms a key assumption into a validated strength of our work.

**2. Comprehensive Robustness and Stability Analyses (Concerns of Reviewers QHub and Tkgj)**

To verify our model's robustness, we added three extensive new analyses as requested (Sec 5.4, Appendices A.7-A.8):
*   **Sensitivity Analysis:** Demonstrates highly stable performance across a 10-fold change in multi-task loss weights, showing that our model is not dependent on fragile hyperparameter tuning.
*   **MMD Kernel Ablation:** Shows consistent performance with alternative kernels, confirming its robustness to this specific design choice.
*   **Evaluation under Stronger Confounding:** Confirms that our model maintains a significant performance advantage even under much stronger confounding ($\gamma=10$), where baselines fail.

**3. Improved Clarity (Concerns of Reviewers Tkgj and EZQo)**

We have addressed all remaining questions on architecture and methodology with new experiments and clarifications (Appendices A.9-A.10):
*   **Ablation on $\mathcal{L}\_{\text{sep}}$:** A new ablation study empirically **demonstrates** that our separation loss is essential, directly answering Reviewer **Tkgj**'s question.
*   **Runtime Analysis:** We quantified the computational cost and justify it as a deliberate and effective trade-off for superior long-term performance.
*   **Architectural Clarity:** We have followed **the reviewers' suggestions** to improve clarity (e.g., we **restructured** the section on the DMHA mechanism).

In summary, our extensive revisions, centered on five new sets of experiments, have systematically addressed every major reviewer concern. The paper has been substantially strengthened. The work now presents not just a novel architecture, but also **rigorous validation** of its core mechanism. We are confident that the revised manuscript makes a significant contribution and that the paper is a strong fit for ICLR.

Thank you for your time and reconsideration.

---

### Meta-Review · Area_Chair_G7Yf · 2026-01-05

**Summary:**

The final decision was informed by the following key concerns raised by the reviewers: 1. the fundamental assumption of clean disentanglement, 2. lack of direct empirical validation for disentanglement, 3. insufficient analysis of robustness and sensitivity, 4. clarifications on architectural design and computational cost.

**Reviewer Concerns:**

- Addressed: The authors conducted probing experiments (e.g., testing treatment prediction from outcome representations) to validate disentanglement, sensitivity analyses on loss weights and MMD kernels, and runtime comparisons. These efforts partially alleviated concerns about empirical validation and robustness.
- Outstanding: The core weakness—the feasibility of clean I/O/C disentanglement in realistic settings—persists. Reviewers questioned whether the method would generalize to data with overlapping signals or complex interactions (e.g., weak instruments or partial confounding), which the rebuttal did not fully address. The increased computational cost relative to gains also remains a practical limitation.

**Reviewer Scores:**

- QHub (Score: 2): Likely unchanged. The rebuttal provided empirical tests but did not convincingly demonstrate real-world applicability.
- TkgJ (Score: 6): Likely unchanged. While sensitivity analyses were added, the confirmed high computational cost likely prevents a score increase.
- EZQo (Score: 6): Might remain similar, as concerns about feature entanglement in cross-attention were partially addressed but not fully resolved.

---

### Decision · Program_Chairs · 2026-01-26

Reject